# Durable Janus membrane with on-demand mode switching fabricated by femtosecond laser

Zehang Cui[1,2,5], Yachao Zhang[1,3,5], Zhicheng Zhang[1], Bingrui Liu[4], Yiyu Chen[1,2], Hao Wu[1], Yuxuan Zhang[1], Zilong Cheng[1], Guoqiang Li[2], Jiale Yong[1], Jiawen Li[1], Dong Wu[1], Jiaru Chu[1] & Yanlei Hu[1] ✉

Despite their notable unidirectional water transport capabilities, Janus membranes are commonly challenged by the fragility of their chemical coatings and the clogging of open microchannels. Here, an on-demand mode-switching strategy is presented to consider the Janus functionality and mechanical durability separately and implement them by simply stretching and releasing the membrane. The stretching Janus mode facilitates unidirectional liquid flow through the hydrophilic micropores-microgrooves channels (PG channels) fabricated by femtosecond laser. The releasing protection mode is designed for the in-situ closure of the PG channels upon encountering external abrasion and impact. The protection mode imparts the Janus membrane robustness to reserve water unidirectional penetration under harsh conditions, such as 2000 cycles mechanical abrasion, 10 days exposure in air and other rigorous tests (sandpaper abrasion, finger rubbing, sand impact and tape peeling). The underlying mechanism of gridded grooves in protecting and enhancing water flow is unveiled. The Janus membrane serves as a fog collector to demonstrate its unwavering mechanical durability in harsh real-world conditions. The presented design strategy could open up new possibilities of Janus membrane in a multitude of applications ranging from multiphase separation devices to fog harvesting and wearable health-monitoring patches.

Janus membranes, which feature opposing wettability on two sides, have garnered significant attention because of their distinctive water directional transmembrane transportation capability[1,2]. Typically, the fabrication process of Janus membranes involves inducing vertical oriented microchannels and applying chemical coatings on one side to achieve opposite surface wettability[3], that is,

laying hydrophilic or hydrophobic coatings on intrinsically hydrophobic or hydrophilic substrates, respectively[4–6]. Water flow could be pumped unidirectionally from the hydrophobic side to the hydrophilic side through the microchannels, while being impeded in the opposite direction. This has enabled rapid advancements in a diverse range of applications such as switchable oil/water

[1]CAS Key Laboratory of Mechanical Behavior and Design of Materials, Key Laboratory of Precision Scientific Instrumentation of Anhui Higher Education Institutes, Department of Precision Machinery and Precision Instrumentation, University of Science and Technology of China, Hefei 230027, China. [2]School of Manufacture Science and Engineering, Key Laboratory of Testing Technology for Manufacturing Process, Ministry of Education, Southwest University of Science and Technology, Mianyang 621010, China. [3]Anhui Province Key Laboratory of Measuring Theory and Precision Instrument, School of Instrument Science and Optoelectronics Engineering, Hefei University of Technology, Hefei 230009, China. [4]Key Laboratory of Agri-Food Safety of Anhui Province, School of Resources and Environment, Anhui Agricultural University, Hefei, Anhui 230036, China. [5]These authors contributed equally: Zehang Cui, Yachao Zhang. ✉e-mail: huyl@ustc.edu.cn

separation[5,7], personal moisture and healthcare management[6,8,9], and sensor devices[2,10].

However, Janus membranes have encountered two common and serious problems in practical applications, leading to the significantly shorter product lifetimes of Janus membranes than expected (Fig. 1). (i) The chemical coatings have poor resistance to mechanical wear[11,12]. Abrasion exposes underlying materials and thus disrupts the local wettability gradient[13]. (ii) When the Janus membrane is in a non-working state, open microchannels are prone to clogging due to the deposition of ambient pollutant particles[4,12,14,15]. Specifically, for intrinsically hydrophobic substrate-based Janus membranes, the lifespan of hydrophilic coatings is exceedingly short (typically <6 h), owning to the failure of hydrophilicity caused by the adsorption of low surface energy substances in air[16,17]. To date, scant attention has been paid to the durability issue of Janus membranes, despite their imperative and urgent need for practical applications in harsh real-world environments.

Take hydrophobic substrate-based Janus membrane as an example, priority must be given to the consideration of hydrophilicity durability. It is known that the surface roughness and the chemical properties synergistically enhance the hydrophilicity of a solid surface[18]. Therefore, the improvement of both the mechanical stability of surface roughness and durability of chemical properties is required. With respect to the robust wettability surfaces, current research efforts are mainly devoted to the hydrophobicity durability[13,19]. Various approaches have been explored, such as the implementation of an open crater-like hard armor to protect the water-repellent surface roughness[20], or the integration of porous bulk microskeleton with hydrophobic nanofiller as a composite membrane to withstand abrasion[21,22]. However, with the wettability microstructure exposed to mechanical wear and ambient pollutant, these strategies are not applicable to the hydrophilicity durability of the hydrophobic substrate-based Janus membrane.

Here, we present a mode switching strategy that alternates between working Janus mode and protection mode to address the aforementioned issues. The hydrophilic micropores connected by orthogonally gridded grooves are processed on a pre-stretched silicone membrane via femtosecond laser microfabrication and hydrophilicity reagent treatment. For working Janus mode, the unidirectional flow of water from the hydrophobic to the hydrophilic side through the micropores and microgrooves channels (PG channels) is allowed. The protection mode is designed to protect the hydrophilic coatings and microstructures of the non-working membrane against external abrasion and impact. By simply releasing the silicone membrane, both the micropores and microgrooves seal to protect themselves. In this manner, the membrane exhibits exceptional resistance to rigorous durability tests, including sandpaper abrasion, finger rubbing, sand impact, and tape peeling, while maintaining its ability of water unidirectional transportation after stretching. Notably, the membrane's functionality remains active even after enduring 2000 cycles of mechanical abrasion and being exposed to air for 10 days. It is revealed that the interconnected grooves play a crucial role, not only in ensuring the durability in protection mode but also in elevating the water unidirectional transportation speed in Janus mode. In the proof-of-concept application, the Janus membrane serves as a fog collector, demonstrating its mechanical stability in harsh real-world environments. This study provides valuable inspiration for designing advanced durable fluid manipulation devices toward a wide range of applications, such as fog collection[23–25], functional clothing[26], open microfluidics and water purification[27–29].

## Results

### Structural design and fabrication

The structural design concept and detailed fabrication process of the robust Janus membrane is shown in Fig. 2a. First, the silicone membrane is equally stretched in both directions. Then femtosecond laser one-step scanning-drilling hybrid fabrication technique is employed to produce orthogonal grid microgrooves on the surface and micropores in the vertical direction. The micropores are designed to locate inside the groove for better concealment (Fig. 2b and Supplementary Fig. 1a). The hydrophilic reagent, composed mainly of hydrophilic polyethylene glycol (PEG), is attached to the pores and grooves (PG channels) via the intermolecular hydrogen bond interactions (Supplementary Fig. 1b). This modification makes the intrinsic hydrophobic PG channels to be hydrophilic (HL, top side), and the opposite side possess intrinsic hydrophobicity of silicone (HB, bottom side). In stretching Janus mode, the top side and bottom side exhibit hydrophilicity (CA ~ 10°) and hydrophobicity (CA ~ 125°), respectively (Fig. 2c). Due to the wettability gradient of the membrane, anti-gravity unidirectional water transportation is realized from the bottom

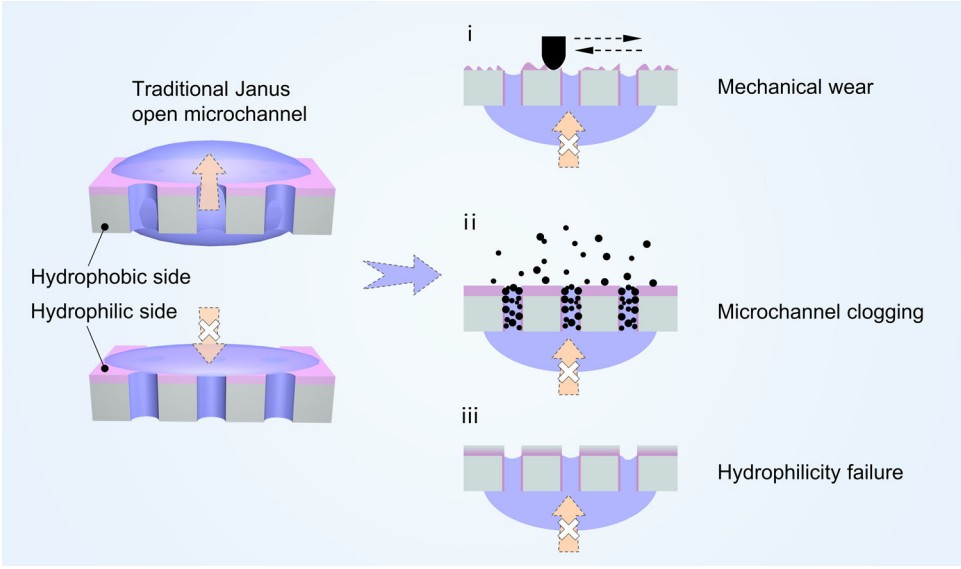

**Fig. 1 | Challenges threaten the actual product lifespan of traditional Janus membranes in practical applications.** i Exposure of underlying materials and disruption of the local wettability gradient when subjected to mechanical wear. ii Open microchannels clogged by the ambient pollutant particles deposition. iii Failure of hydrophilic coatings caused by the low surface energy substances adsorption in air.

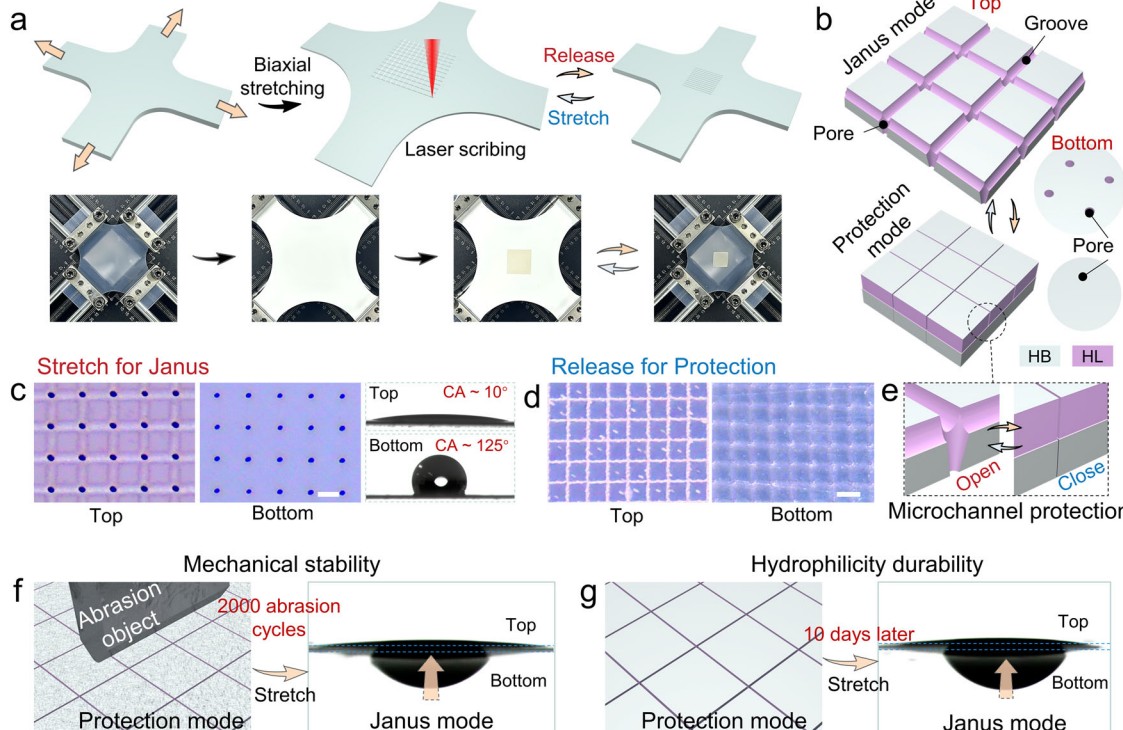

**Fig. 2 | Design and fabrication of the robust Janus membrane. a** Schematic and experimental images of femtosecond laser one-step scanning-drilling hybrid fabrication on a biaxially stretched silicone membrane. **b** Schematic diagram of the switching between the stretching Janus mode and releasing protection mode, and the corresponding microstructure on the top and bottom sides, respectively. **c** Optical microscope images of top PG channels side, bottom micropores side, and the corresponding water contact angles in Janus mode. **d** After releasing to

protection mode, all the PG channels on the top side and the micropores on the bottom are closed. Scale bars: 100 μm. **e** Close-up diagram showing that the reversible opening and closing of the PG channels imparts the membrane Janus function and protection, respectively. In the protection mode **f** the hydrophilic PG channels can resist 2000 cycles of mechanical abrasion and **g** exposure in air for a duration of 10 days, while the water unidirectional transportation ability remains unaffected.

hydrophobic side to the top hydrophilic side. To resist mechanical abrasion and impact, the protection mode can be achieved by simply releasing the membrane (Fig. 2b, d). In the protection mode, the microgrooves and micropores are fully closed, and liquid cannot flow through the channels (Fig. 2e, Supplementary Movie 1). During the durability tests, only the matrix material is exposed to abrasion and impact which acts as "soft armor", thus both the hydrophilic coating and microstructures are well preserved. For the corresponding CAs of the protection mode, the top side changes to hydrophobicity because of the concealment of hydrophilic PG channels (Supplementary Fig. 1c). Remarkably, the protection mode can even resist 2000 mechanical abrasion cycles and 10 days of air exposure, preserving unidirectional water transport through PG channels in Janus mode, indicating outstanding mechanical stability and hydrophilicity durability (Fig. 2f, g).

### Characterization of structure and water unidirectional penetration

By releasing and re-stretching the membrane, the structure can be regulated in a controllable manner. The morphologies and profiles of the structures at different strains are obtained by using a confocal laser scanning microscopy (CLSM) (Fig. 3a, b). The strain value ($\varepsilon$) is calculated according to the equation $\varepsilon = (L-L_0)/L_0$, where $L$ and $L_0$ are the membrane length at the tensile state and the initial state, respectively (Supplementary Fig. 2). Typical 3D CLSM images of PG channels at strain values of 20%, 60% and 120% are shown in Fig. 3c. When the strain is 120%, the width of groove measures 110 μm. As the strain decreases, the groove width gradually decreases until it returns to a flat state at the strain of zero. Correspondingly, the pore spacing is reduced from 200 μm at $\varepsilon = 120\%$ to 100 μm at $\varepsilon = 0$, and the pore size

is changed from 35 μm to complete closure (Fig. 3d). The depth of the groove increases from 80 μm at $\varepsilon = 120\%$ to 100 μm at $\varepsilon = 40\%$, and then changes to be immeasurable due to the closure of the groove (Supplementary Fig. 3). In particular, both the micropores and microgrooves exhibit a closing state as the applied strain is completely released ($\varepsilon = 0$), namely, the protection mode.

The water unidirectional penetration time of the structure at different strain values is investigated as shown Fig. 3e. In protection mode, the micropores and microgrooves are closed, preventing water from flowing through the microchannels (Supplementary Fig. 4, Supplementary Movie 1). When the applied strain increases from 20% to 80%, there is a significant decrease in penetration time from 114.5 s to 3.8 s, whereas it remains relatively stable within the strain range of 80% to 120%. Hence, the Janus mode with a stretching strain of 80% is utilized in the subsequent experiments and discussion. To characterize the asymmetric wettability, the apparent dynamic contact angles on each side of the Janus membrane are measured. On the top side, a rapid dynamic decrease in the contact angle is observed from initially ~38.2° to ~4.9° within 1.0 s, indicating the hydrophilicity of the PG channels. In contrast, the bottom side exhibits distinct wetting characteristics. The water droplet with an initial contact angle of ~123.8° do not spread laterally at the bottom. But the contact angle also gradually decreases due to the water droplet permeates through the hydrophilic PG channels to the other side (Fig. 3f, Supplementary Fig. 5). In addition, to further evaluate the functionality and durability, the prepared membrane is repeatedly stretched and released for 20 cycles, and the penetration time is measured after each cycle (Fig. 3g). The results demonstrate the robust switchability of the membrane between Janus and the protection mode. The slight increase in penetration time observed during repeated cycles is attributed to the consumption of

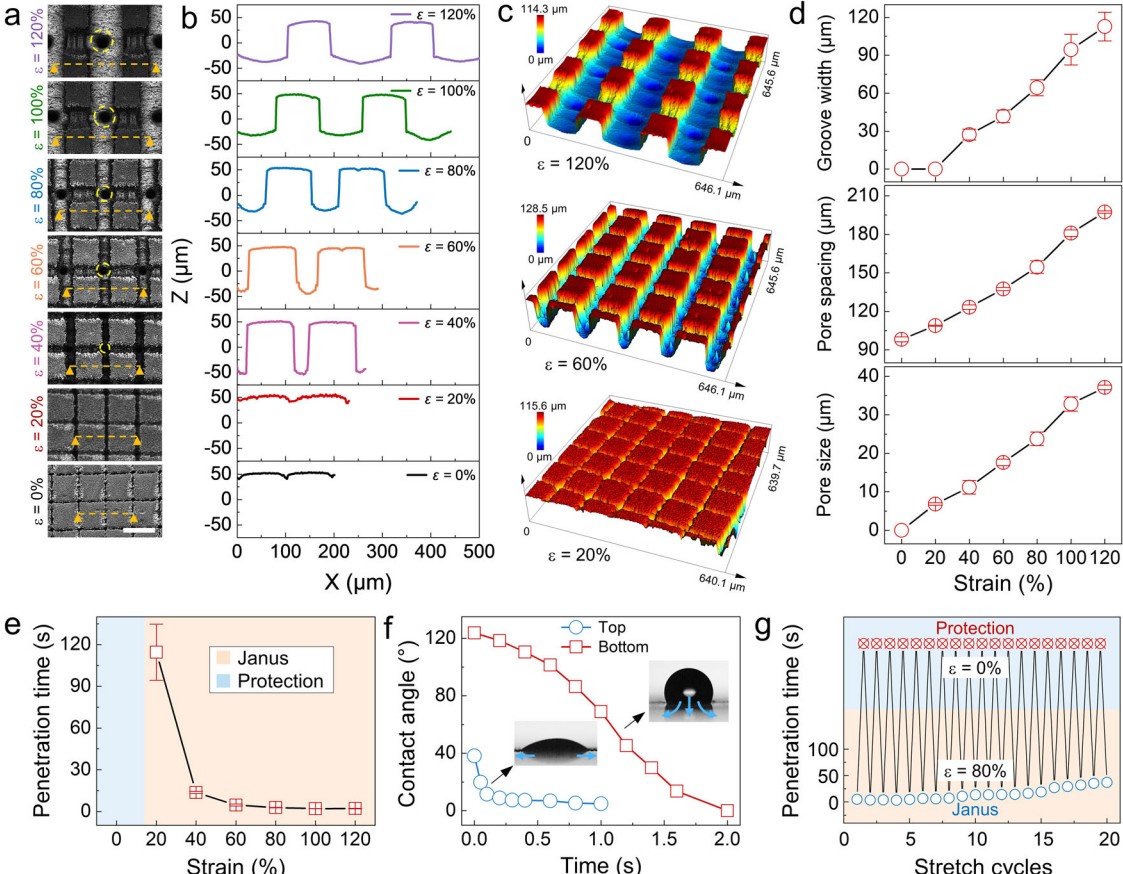

**Fig. 3 | Structural and water unidirectional penetration characterization of the prepared membrane. a** Optical images of PG channels and **b** corresponding surface profiles under different applied strains. Scale bar: 100 μm. **c** Representative CLSM images of the PG channels at strain values of 20%, 60% and 120%, respectively. **d** Variations of pore size, pore spacing and groove width under different strains. **e** Variation of water unidirectional penetration time from the bottom to the top side of the Janus membrane at different stretching strains. **f** Dynamic apparent contact angles of water dropped on the hydrophilic and hydrophobic sides. **g** The robust water unidirectional penetration as continuously switching between the Janus mode and the protection mode over 20 cycles. The error bars represent the standard deviation of three independent measurements. Source data are provided as a Source Data file.

hydrophilic coating (Supplementary Fig. 6-7). Considering that cracks or defects may arise in microgrooves during the stretching process, potentially compromising the water unidirectional penetration function, we investigate the impact of repeated stretching on the Janus membrane function. Results show that new cracks only appear during the first three stretch cycles (Supplementary Fig. 8). To mitigate the effects of these cracks on Janus membrane performance, a repeated hydrophilic modification approach is employed. Compared with the water unidirectional penetration function of Janus membrane modified one time in stretched state (renders ineffective after just 7 stretching/releasing cycles, Supplementary Fig. 9), the robust switchability observed over 20 cycles indicates that the repeated stretching during the modification process eliminates the impact of microgrooves surface cracks on Janus membrane performance.

## Mechanical durability and water unidirectional penetration mechanism analysis

The long-term mechanical durability of the prepared Janus membrane is systematically investigated (Fig. 4a, Supplementary Fig. 10, Supplementary Movie 2). To see more details, we examine the morphologies and profiles of the PG channels before and after long-term mechanical abrasion (Fig. 4b, c). After 2000 abrasion cycles, the interior of the microgrooves remains closed in protection mode. For the Janus mode, the groove depth decreases because of the mechanical abrasion. As the schematic diagram shown in Fig. 4d, although the outermost layer of the silicone membrane is

abraded, the structural integrity and hydrophilicity of the closed PG channels inside the matrix remain unaffected. After re-stretching to the Janus mode, the membrane maintains its water unidirectional penetration even after 2000 abrasion cycles, demonstrating the effective protection of the hydrophilic PG channels (Supplementary Movie 3). To understand the impact of mechanical abrasion on the water unidirectional penetration performance, we measure the abrasion depth and the droplet unidirectional transportation time with the membrane samples subjected to different abrasion cycles (0, 500, 1000, 1500 and 2000 cycles), as shown in Fig. 4e. The abrasion depth is defined by the profile height difference between the worn area and original surface. The morphologies and profiles can be observed in Supplementary Fig. 11. With the increasing of the abrasion cycles, the abrasion depth increases to 42 μm, and the water unidirectional penetration function remains active with a slight increasing penetration time from 3.8 s to 6.5 s. Compared with the Janus membrane in open grooves and pores state with no protection, the water unidirectional penetration function fails after only 300 abrasion cycles, indicating the protection mode imparts Janus membrane outstanding mechanical stability. In addition, the membrane is exposed to air for 10 days in protection mode, the water unidirectional penetration time only increases from 3.8 s to 4.3 s, demonstrating that the protection mode can effectively prevent contamination from the low-surface-energy organic molecules in the air and improve the durability of hydrophilicity of PG channels. As a comparison, traditional open microchannels with

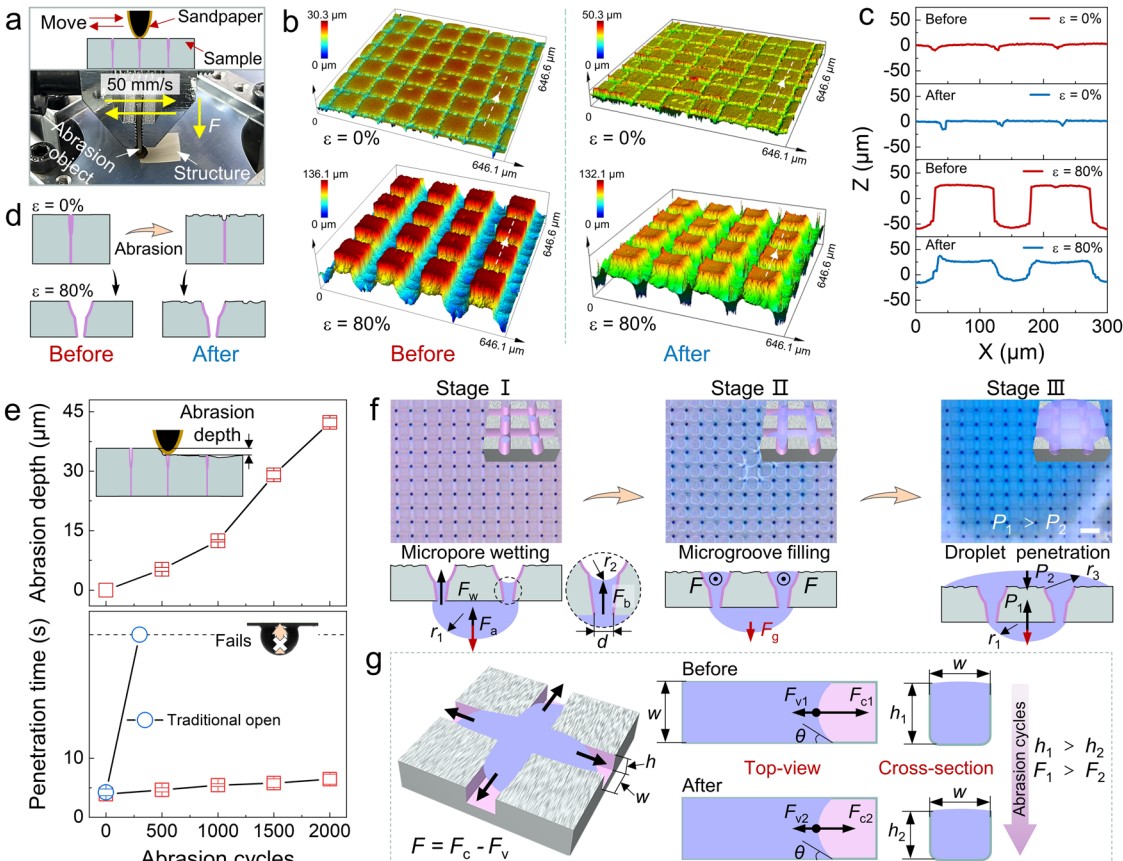

**Fig. 4 | Mechanical durability and water transportation mechanism analysis of the Janus membrane. a** Schematic illustration and optical photograph of the liner abrasion setup. **b** CLSM images and **c** profile comparison of the PG channels in the Janus and protection modes before and after 2000 abrasion cycles, respectively. **d** Schematic diagram of the protection mode enhancing durability of hydrophilic PG channels. **e** The influence of liner abrasion cycles on abrasion depth and water unidirectional penetration time. **f** Microscopic images and force analysis of water unidirectional penetration and spreading on the top hydrophilic PG channels side. Scale bar: 200 μm. **g** The dominant driving force analysis of water flow in microgrooves with increasing abrasion cycles. The error bars represent the standard deviation of three independent measurements. Source data are provided as a Source Data file.

hydrophilic coating of polymer-based Janus membrane change to hydrophobicity under only 6 h of air exposure, and water cannot penetrate from the bottom to the top side (Supplementary Fig. 12).

To gain a deep understanding of the robust unidirectional liquid transportation mechanism of the Janus membrane, microscopic images of water spreading process along the PG channels are captured by a digital camera. In-situ observation of the water spreading behavior on the top side of the Janus membrane shows that the unidirectional penetration transport can be divided into three stages. Water is first pumped into the micropores and wet them (Stage I). Subsequently, water flows through PG channels until it fills all the microgrooves and forms a thin film on the hydrophilic side (Stage II). Finally, a convex droplet emerges rapidly and spreads on the top side (Stage III) (Supplementary Fig. 13a).

In addition, the dominant forces in the whole droplet anti-gravity unidirectional penetration process are carefully analyzed, as shown in Fig. 4f. The gravity of the droplet that has not yet entered the micropore $F_g$ acts a resistance in the vertical advancing direction. It tends to decrease from Stage I to Stage III as the liquid being continuously transported to the top surface. As for the driving force, in Stage I, the wetting gradient force $F_w$, derived from the hydrophilicity gradient in the vertical direction[30–32], the surface tension of the water ($F_a$) and the capillary force within the laser drilling asymmetric conical micropores ($F_b$), collectively drive the droplet from the hydrophobic bottom side into the pore microchannels[4]. And the driving forces can be described as follows:

$$F_w = \gamma l \left( \cos \theta_a - \cos \theta_r \right) \quad (1)$$

$$F_a = \frac{\pi \gamma d^2}{2 r_1} \quad (2)$$

$$F_b = 2 \pi \gamma r_2 \cos \left| \theta + \frac{\alpha}{2} \right| \quad (3)$$

where $\gamma$ is the surface tension of water, $l$ is the effective contact length of the droplets, $\theta_a$ and $\theta_r$ are the local advancing and receding contact angles, $d$ is diameter of the micropore, $r_1$ is the radius of the water droplet, $r_2$ is the radius of the three-phase contact line inside the micropore, $\theta$ is the water contact angle, and $\alpha$ is the taper angle of micropore, respectively. And the magnitude of driving force $F_w$, $F_a$ and $F_b$ can be supposed to remain constant before and after abrasion because of micropores preservation in protection mode.

As the water flows in these microgrooves during Stage II, the driving force $F$ can be described as follows:

$$F = F_c - F_v \quad (4)$$

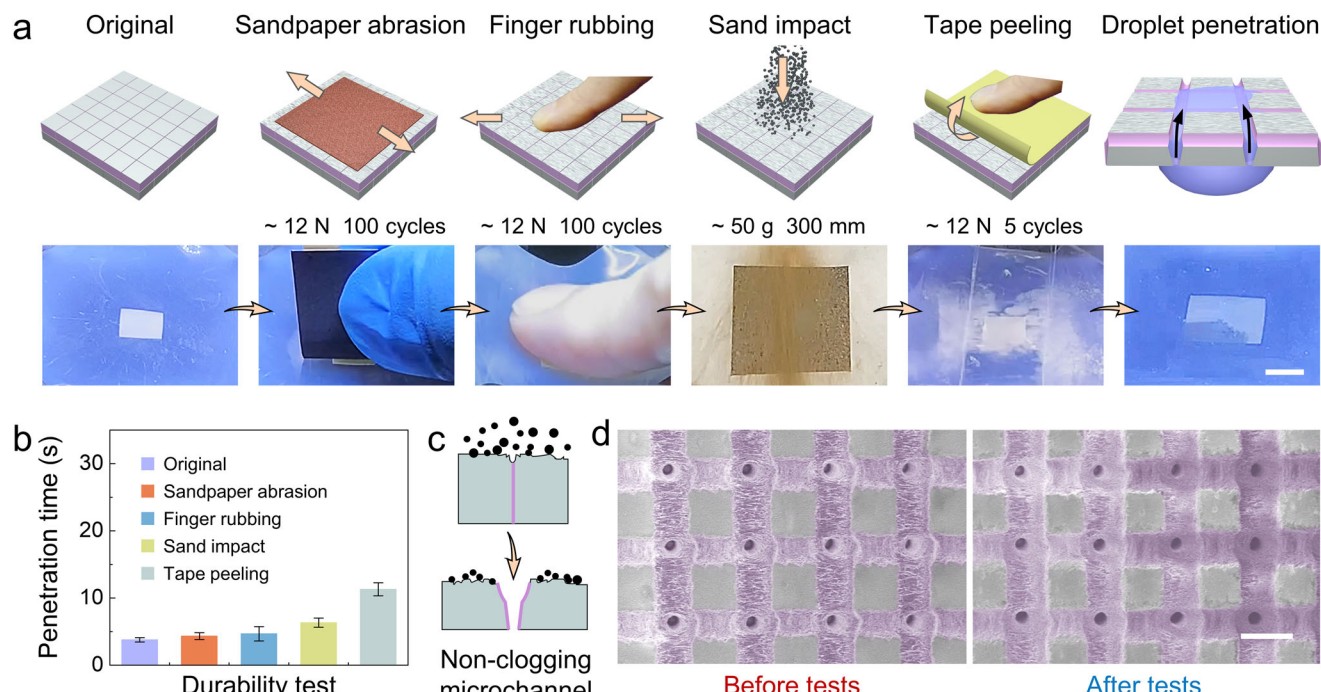

**Fig. 5 | Mechanical durability tests in harsh conditions. a** Schematic and experimental results of sandpaper abrasion, finger rubbing, sand impact, tape peeling tests and the final verification of water unidirectional penetration. Scale bar: 10 mm. **b** Variation of water unidirectional penetration time after each test in (**a**). **c** Schematic illustration of the protection mode imparting the Janus membrane with non-clogging advantage. **d** Pseudo-color SEM images of PG channels (the purple part) before and after all the durability tests. Two membrane samples with the same processing parameters are employed before and after the tests for SEM characterization. Scale bar: 100 μm. The error bars represent the standard deviation of three independent measurements. Source data are provided as a Source Data file.

---

where $F_c$ is the capillary force[33–35], $F_v$ is the viscous force[36,37]. In this equation:

$$F_c = 2\gamma h \cos\theta \tag{5}$$

$$F_v = \frac{3\eta xu}{\varepsilon\zeta(\varepsilon)} \tag{6}$$

where $h$ is the groove depth, $\eta$ is the water viscosity, $x$ is the water transport distance inside the microgroove, $u$ is the flow velocity, and the aspect ratio of the microgrooves is $\varepsilon = h/w$ ($w$ is the groove width). As depicted in Fig. 4g, the groove depths before and after abrasion are donated as $h_1$ and $h_2$, respectively. The values of $F_{c1}$, $F_{v1}$ and $F_{c2}$, $F_{v2}$ (before and after 2000 abrasion cycles) are estimated to be $1.17 \times 10^{-5}$ N, $1.09 \times 10^{-7}$ N and $6.16 \times 10^{-6}$ N, $4.91 \times 10^{-8}$ N, respectively (Supplementary Note 1). Hence, the capillary force is the main driving force. The driving force $F$ is mainly determined by $h$, and $h_1 > h_2$, $F_1 > F_2$ with increasing abrasion cycles. The observed unidirectional penetration time rise in Fig. 4e verifies the reduced driving force caused by the increasing mechanical abrasion cycles. When complete erosion of the microgrooves occurs after about 6000 abrasion cycles, namely $h \sim 0$, water droplets cannot penetrate from the bottom to the top side (Supplementary Fig. 14).

In Stage III, the unidirectional droplet penetration is driven by the Laplace pressure difference $\Delta P$ between the droplets on the top and bottom side[38,39]:

$$\Delta P = P_1 - P_2 = 2\gamma\left(\frac{1}{r_1} - \frac{1}{r_3}\right) > 0 \tag{7}$$

where $r_3$ is the local radius of droplet on the top side, and $r_1 < r_3$. More importantly, in addition to protecting the surface roughness and hydrophilicity inside PG channels, the hydrophilic grooves play a significant role in enhancing the droplet transport velocity. The unidirectional flow time of water droplets (with the same volume of 5 μL) passing through the traditional micropore and the PG channels here is 10 s and 3.8 s, respectively (Supplementary Fig. 13, Supplementary Movie 4). The mechanism of the transportation enhancement is elucidated in Supplementary Fig. 15.

## Rigorous mechanical durability tests

To further characterize the durability of the prepared Janus membrane, more rigorous tests are conducted under harsh conditions. Sandpaper abrasion, finger rubbing, sand impact and tape peeling tests are carried out successively (Fig. 5a, Supplementary Movie 5–9). After each test, the membrane is found to maintain water unidirectional penetration function in Janus mode, and the water penetration time is recorded. From the original (before test) to the final state (after tape peeling), the penetration time increases from 3.8 s to 11.3 s (Fig. 5b, considering the hydrophilic coating consumption). During these tests, the microchannels is closed in protection mode and prevents the pollutant particles from intruding into the PG channels, demonstrating that the non-clogging microchannels are suitable for applications in harsh environments (Fig. 5c). And the pseudo-color SEM images of the PG channels before and after all the tests are displayed in Fig. 5d. Both the pores and grooves (the purple part) remain unscathed after the harsh durability tests. We further perform a comparative experiment in which the above mechanical tests are conducted in Janus mode. Any of the four tests invalidates water unidirectional penetration function, verifying the necessity and importance of the protection mode (Supplementary Fig. 16).

## Durable fog collection

In the real-world harsh condition like the arid desert, the fog collector faces a significant challenge from sandstorm abrasion, during which, the hydrophilic coating is worn and the microchannels are clogged. As

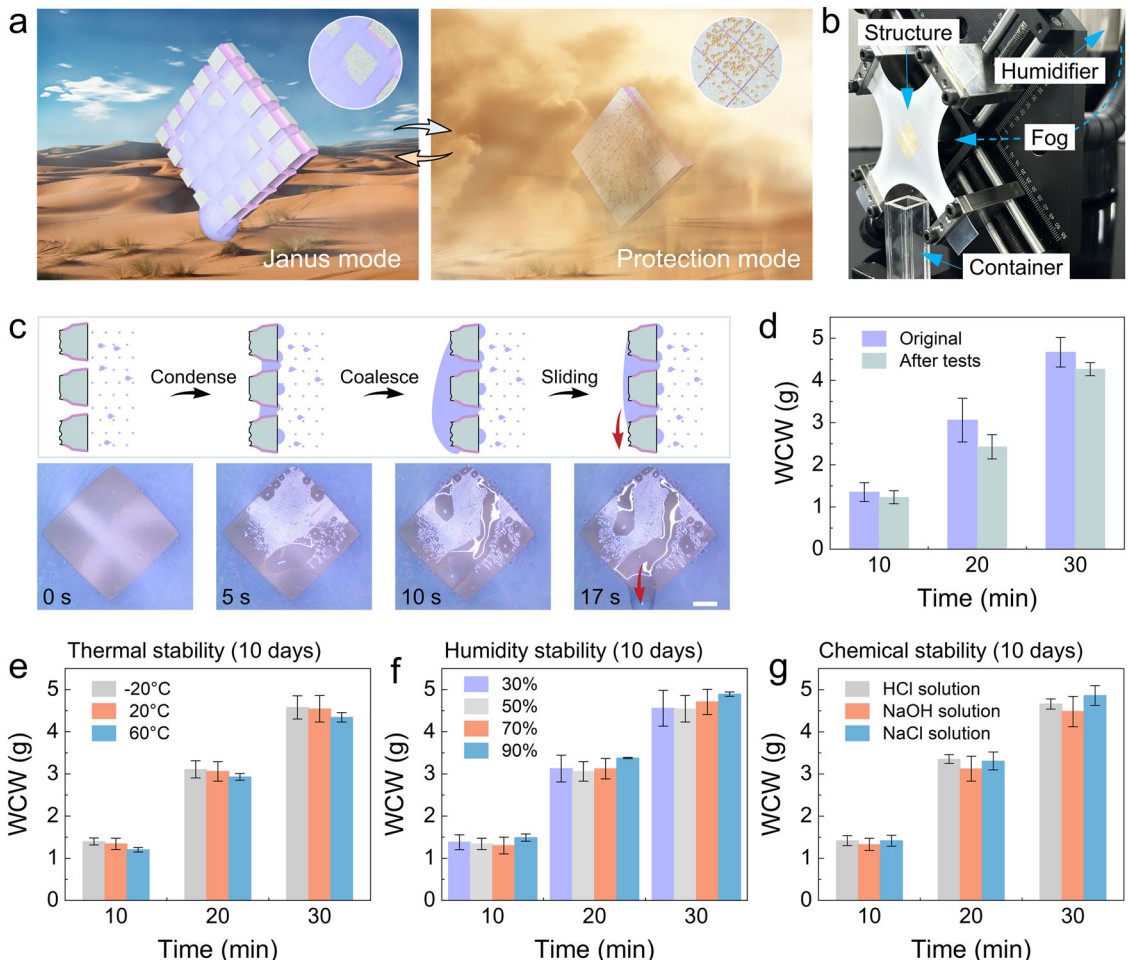

**Fig. 6 | Proof-of-concept durable fog-collection application of the robust Janus membrane. a** Schematic illustration of the durable Janus membrane with on-demand mode switching strategy for fog collection in arid deserts. **b** Optical photograph of the fog-collection system. **c** Schematic and in-situ observation of fog condensation, water drop coalescence, and directional sliding on the top PG channels surface after all the harsh tests. Scale bar: 5 mm. **d** Comparison of WCW of the Janus membrane before and after all the tests during a period of 30 minutes. **e** The WCW changes in 30 minutes of the Janus membranes after 10 days of exposure at a temperature of −20 °C, 20 °C and 60 °C. **f** The WCW changes in 30 minutes of the Janus membranes after 10 days of exposure at a relative humidity of 30%, 50%, 70% and 90%. **g** The WCW changes in 30 minutes of the Janus membranes after 10 days immersion in HCl solution (pH ~ 4), NaOH solution (pH ~ 10) and 5 wt.% NaCl solution. The error bars represent the standard deviation of three independent measurements. Source data are provided as a Source Data file.

an alternative solution, the robust Janus membrane here with mode switching capability can be utilized as a durable fog collector. Figure 6a illustrates a new operation mechanism adaptable to varying time and environmental conditions. During the early morning, the stretching Janus mode efficiently captures overnight accumulated fog. As the fog disperses and sandstorms approach at noon, the protection mode becomes active, providing resistance against mechanical sandy abrasion and preventing microchannel clogging. Additionally, the protection mode can serve as a long-term storage mode, thereby further enhancing the overall durability of fog collectors. The experimental apparatus is shown in Fig. 6b and Supplementary Fig. 17. Fog flows to the bottom side of Janus membrane through the hole behind the clamp. The fog-collecting capability of Janus membrane before and after the above-mentioned durability tests are investigated. To investigate the fog-collecting behavior, a digital camera is used for in-situ observation of the whole process. The schematic diagram in Fig. 6c displays a representative and enlarged scheme illustrating the fog-collecting process, including droplet condensation, coalescence and sliding down. Water droplets sliding down for collection occurs at 15 s and 17 s for Janus membrane before and after the durability tests, respectively (Fig. 6c, Supplementary Fig. 18, Supplementary Movie 10). The results indicate that after all the harsh tests, the Janus membrane preserves outstanding fog collecting performance. The weight of

collected water (WCW) of the Janus membrane for 30 minutes before and after tests are quantitatively compared, and only a 10% water collection decrease occurs after all the harsh tests (Fig. 6d). The comparison of fog collecting performance shows that the fragile hydrophilic PG channels are well protected.

Considering the complexity of the practical application environment of the fog collector, the further long-term thermal, humidity and chemical stability performance of the Janus membrane in protection mode are also investigated. In terms of thermal stability, membranes are exposed in the temperature of −20 °C, 20 °C and 60 °C for 10 days considering the temperature difference between day and night in the desert. Compared with the original state (the freshly prepared Janus membrane) in Fig. 6d, the WCW of these membranes for 30 minutes remains largely unchanged, as shown in Fig. 6e. In addition, membranes are kept in the conditions with different relative humidity (30%, 50%, 70% and 90%) and immersed into different chemical solutions (HCl, NaOH and NaCl). As shown in Fig. 6f, g, after 10 days' treatment, the WCW of a period 30 minutes also keep consistent compared with the original state. The results indicate the outstanding thermal, humidity and chemical stability of the Janus membrane. What is worth noting that the core novelty of this work lies in the proposal of the mode switching strategy that can significantly enhance the mechanical durability of the Janus membrane regardless of the specific type or

composition of the hydrophilic coating. Here, an alternative PVA/silica hydrophilic coating is introduced for HCl and NaOH solution immersion tests. We can expect that this not only demonstrates the universal hydrophilic coating protection effect of the mode switching strategy, but also broadens the range of options for hydrophilic coatings that are both mechanically and chemically durable.

## Discussion

In order to tackle the common and severe issue of Janus membrane function failure in real-world applications, an on-demand mode switching strategy is proposed in this study. The strategy of considering working and protection modes separately, implemented by stretching and releasing the membrane. The stretching Janus mode enables water unidirectional penetration, and the releasing protection mode resists abrasion and impact. With the on-demand switching between the two modes, the durability of Janus membrane has been significantly enhanced, water unidirectional penetration could be reserved under harsh mechanical impacts. In the proof-of-concept application, the Janus membrane serves as a fog collector, demonstrating its mechanical stability in harsh real-world environments.

While the releasing protective mode can significantly enhance the durability of the Janus membrane in the face of external abrasion and impact, as well as during long-term storage, future work may focus on improving the durability of the flexible Janus membrane in its active working mode. Improving the longevity of the hydrophilic coating would further enhance the robustness of the device in working mode. Additionally, the wear resistance of the silicone has not been optimized. For instance, one improvement could involve incorporating carbon nanotubes into the silicone to enhance its wear resistance in protective mode. In the future, a comprehensive enhancement of device durability could be achieved by combining a durable hydrophilic coating design with materials that offer both wear resistance and elasticity. We envision that the presented design strategy has the potential to advance Janus membranes toward a myriad of practical applications, such as multiphase separation and purification, controllable fluid manipulators, and wearable health monitoring patches.

## Methods
### Material fabrication
Silicone membranes with a thickness of 500 μm (purchased from Pureshi Co. Ltd.) were cut into a symmetrical cruciform shape and stretched utilizing a homemade biaxial stretching clamp. A regenerative amplified Ti:sapphire femtosecond laser system (Legend Elite-1K-HE, Coherent, USA) with a 104 fs pulse width, 1 kHz repetition rate and 800 nm central wavelength was employed in the PG channels fabrication. The laser beam was guided on the stretched silicone membrane surface by a 2D galvoscanner system (SCANLAB, Germany). Femtosecond laser one-step scanning-drilling hybrid fabrication involving orthogonally crossed line-by-line consecutive scanning to generate gridded microgrooves and laser microdrilling to create uniform micropores[40–43]. An optimized laser power of 50 mW, scanning speed of 50 mm/s, scanning times of 15 and marking time of 100 ms were adopted. The hydrophilic reagent Mesophilic-2000 (hydrophilic PEG solution, MesoBioSys. Co. Ltd., Wuhan, China) was used in the sample surface hydrophilic modification. The reagent was applied on the stretched surface by cotton swabs, and the modified sample was placed at room temperature for 5 min to evaporate the solvent. The membrane was released and restretched three times for hydrophilic modification to obtain the hydrophilic top surface of the Janus membrane.

### Characterizations
All contact angle measurements were conducted using a CA100C machine (Innuo, China) at ambient temperature. The CA value was obtained by averaging at least three measurements performed at different positions on the same sample. Microdroplets were generated by a syringe and placed at the bottom of the Janus membrane, and the water unidirectional penetration process was recorded by a CCD of the contact-angle system. An OLS5000 confocal laser scanning microscopy was used to capture the surface 3D microstructure topography profiles. The top-view images of PG channels were captured by a CCD (Mindvision) equipped with an optical lens. The water droplets dynamic contact angle process was recorded by a high-speed camera (CHRONOS 2.1-HD, Canada). Due to the size of the stretching stage was larger than that of the SEM sample room, all the biaxially stretched membranes in experiments with structures were fixed on hard plastic plates by screws, and then the membranes were cut off for SEM characterization (EVO18, ZEISS, Germany). The FTIR spectra of the samples were recorded using a Nicolet iN10 Fourier Transform Infrared Microscope (Thermo Scientific Instrument Co. U.S.A).

### Mechanical durability tests
The long-term mechanical abrasion test of the Janus membrane was conducted on a homemade apparatus, which comprises a vertically placed force sensor with an abrasion probe, and an electrically controlled linear slide platform mounted with a z-axis stage. The membrane in protection mode was mounted on the electrically controlled linear slide platform. A polymer probe covered with sandpaper was pressed against the membrane with a defined vertical pressure of 40 kPa. The long-time abrasion test was then performed by reciprocating the probe at a constant velocity of 50 mm/s. For the sand abrasion test, 50 g of commercial desert sand (purchased from Tianjingsha Co. Ltd.) impacted the surface from a 300 mm height, while the silicone membrane was held at 45° to the horizontal surface. The force applied in sandpaper abrasion, finger rubbing and tape peeling on the sample was measured by a precise force sensor under the stage (HZC-T, Chengying Sensor Co. Ltd. Bengbu, China), and the average pressure was measured -12 N with multiple mechanical cycles.

### Thermal, humidity and chemical stability tests
Janus membranes in protection mode were subjected to various conditions, including different temperature, humidity and chemical solutions, to investigate their long-term stability. After 10 days' treatment, the membranes were stretched to Janus mode and the weight of collected water for 30 minutes was quantitatively compared with the original state (the freshly prepared Janus membranes) to evaluate the stability. Specifically, the thermal stability was tested by keeping the samples in refrigerator (−20 °C), room condition (20 °C) and oven (60 °C). For humidity stability assessment, the membranes were exposed to homemade chambers with humidity levels of 30%, 50%, 70%, and 90%, respectively. In chemical stability tests, membranes were immersed into chemical solution of HCl (pH ~ 4), NaOH (pH ~ 10) and NaCl (5 wt.%), respectively. A hydrophilic coating, consisting of PVA/Silica solution[10], was uniformly sprayed on the top side of Janus membrane by using a spray gun at a distance of 10 cm for chemical solution HCl and NaOH immersion tests.

## Data availability
All data needed to evaluate the conclusions in the paper are present in the manuscript and Supplementary Information. The data are also available upon request from the corresponding author. Source data are provided with this paper.

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

## Acknowledgements

This work was supported by the National Natural Science Foundation of China (Nos. 52122511, 52105492, 62325507, 52375582, 61927814, U20A20290), Major Scientific and Technological Projects in Anhui Province (202203a05020014), Youth Innovation Promotion Association CAS (Y2021118) and Fundamental Research Funds for the Central Universities (WK5290000004). We acknowledge the Experimental Center of Engineering and Material Sciences at USTC for the fabrication and characterization of samples. This work was partly carried out at the USTC Center for Micro and Nanoscale Research and Fabrication.

## Author contributions

Z.C. (Zehang Cui), Y.Z. (Yachao Zhang) and Y.H. conceived the idea and designed the experiments. Z.C. (Zehang Cui), Y.Z. (Yachao Zhang), Y.C., Y.Z. (Yuxuan Zhang) and Z.C. (Zilong Cheng) performed the experiments. Z.Z. and B.L. participated the chemical composition analysis and characterization. H.W., G.L., J.Y. and J.L. provided constructive suggestions for the results and discussion. Z.C. (Zehang Cui), Y.Z. (Yachao

Zhang) and Y.H. wrote and revised the paper. Y.H., D.W. and J.C. supervised the project.

## Competing interests

The authors declare no competing interests.
