## [Peer Review File · Nature Communications]

REVIEWER COMMENTS

Reviewer #2 (Remarks to the Author):

In the submitted manuscript Zehang Cui et. al. fabricated Janus membranes using femtosecond laser microfabrication on pre-stretched silicone membrane surface. Specifically, hexagonal grooves were inscribed with help of laser on stretched silicone surface with pores drilled inside the grooves. The top hydrophilic surface was created by reagent treatment whereas inherent hydrophobicity of the silicone at the bottom was utilized to generate Janus membranes. The mechanical stability and performance of the membrane is evaluated under extreme harsh conditions and no loss in performance was observed. Although the substrates are well-characterized and results adequately supported by experimental studies, due to (i) lack of basic correlation between material studied and application (ii) lack of discussion in support of observation and (iii) overall novelty of work, I do not support the publication of this manuscript in a reputed journal like Nature Communication with high impact factor. Please see below my concerns:

1. The details about Mesobiosys, a reagent used for hydrophilization are missing. Basically, authors did not mention how this makes the surface hydrophilic? What is the chemistry driving the interaction between silicone substrate and mesobiobiosys? Failing to provide the details limits the complete understanding of chemistry driving the substrate performance.
2. The technique of stretching and creating patterns using femtosecond laser for surface modification is well known and reported multiple times earlier, for example: <https://doi.org/10.1021/acscami.1c02121>. My concern is stretching silicone surface and patterning it then releasing it to open and close the pores is far from real-life application. The authors are good at patterning the surface but where these stretched Silicone substrates used for fog collection in reality? Much simpler designs are proposed and demonstrated earlier example: <https://doi.org/10.1021/acs.langmuir.2c02697>
3. Mechanical abrasions are studied here and its good to preserve the microstructure and pores but this may not be necessary/needed for a fog collector. Other application like smart clothing maybe more relevant.
4. Some redundant tests are carried out after which the performance did not change, to me this is not a surprise. For example (1) release of strain reduced the pore size from 35 μm to zero. This closure state is called “protective mode” and this phenomenon is as expected. (2) No water flow is observed in closed state and under applied strain state, again quite expected. (3) Silicone membranes are elastomer in nature, and I did not see any novelty that they can be stretched for 20 cycles without loss in performance.
5. Change in penetration time after stretching cycles (Fig 6 in SI), how to authors support this observation? Merely stating “consumption of hydrophilic coating” does not suffice. Can authors quantify the results in term of consumed reagent? Or was this observation due to change in the

intrinsic material properties of silicone like cracks/defect generated? Was hydrophilicity generated o the reagent used bound to silicone covalently/ionically/supramolecularly/van der waals attraction?

6. In line 136/137 authors declare “In real-world applications, membranes are usually subjected to harsh conditions such as contact abrasion” were are such conditions? Can authors specify such applications with respect to current studies where this “protective mode” can be utilized specifically w.r.t silicone membranes?

7. During the 2000 cycle abrasion test, the groove depth decreased from 82 μm to 43 μm and groove and the no significant performance change was observed (unidirectional water flow behavior). But this is merely because the test was designed in a manner such that complete erosion of grooves does not take place. What is missing is insight on water transport properties, was there a significant change in resistance to water flow?

8. What are the properties of Janus polymer used for comparative studies (Supplementary Fig. 9). Without providing the membrane characteristic like thickness, pore size, contact angle at hydrophilic and hydrophobic surface etc. such tests will be inaccurate/incomplete.

9. The magnitude of driving force is expected to remain constant before and after abrasion because the hydrophilic surface was protected during abrasion and the active thickness available for water flow is almost constant.

10. During rigorous durability tests and fog-collection application test, efforts are dedicated to mechanical stability. How about the performance under different relative humidity conditions for long-term? Can here, the tests should be performed under two conditions (1) protected morphology and (2) open pore morphology? This will inform about the overall stability the designed membrane combined with the coatings and physical attributes.

11. Study summarized in figure 12 show importance of the protective mode, but this may be impractical in realistic environment.

12. Sandstrom is a specific case that supports this study, other environmental factors like change in temperature, humidity (mentioned above) are not studied which makes it difficult to visualize the applicability of this material under realistic conditions.

Reviewer #3 (Remarks to the Author):

Major Revision

The author has demonstrated the feasibility of a Janus membrane in a multitude of applications ranging from multiphase separation devices to fog harvesting and wearable health-monitoring patches. Overall, this is a decent research topic which can be noteworthy for researchers' studying

Janus membrane fabrication and separation technology. However, in this research paper, there are some important queries which should be addressed:

1. Starting with the abstract, kindly highlight the best outcome of the fabricated Janus membrane for better understanding.
2. Very importantly, the generation of hydrophilic pores via femtosecond laser must be well in terms of high scientific discussion. The mechanism must be included.
3. The objective of this research study is in premature stage and must be well elaborated (especially last paragraph).
4. What can we gain from Figure 4 (e)? Please come up with valid references for better readability. The increased penetration time is almost twice as the original time. Thus, the mechanical stability seems to be bit uncertain.
5. Can authors explain the chemical durability of this fabricated Janus membrane? This aspect can enhance the quality of this manuscript.

Reviewer #4 (Remarks to the Author):

The manuscript discusses a mode-switching strategy to protect Janus membranes through stretching and releasing. The work described is novel and emphasizes the long-term stability and mechanical durability of membranes, an area which is often overlooked in membrane development. However, the manuscript is incomplete and poorly structured in its current form. I suggest the manuscript be accepted after some revision.

- To capture the membrane's ability for fog collection, the authors are advised to conduct thermogravimetric analysis to quantify water absorption and moisture content with varying strain %, and to discuss the factors that influence the same.
- The 'discussion' section in its current form is not a discussion, but a conclusion.
- Much of the results' section also include methods of how measurements were taken, which should be limited to the methods section. As an example, lines 140-142 explain how the sandpaper test was carried out. This does not belong in 'results'. All methods must be moved to 'methods'.

- As a result of the above, the 'methods' section is weak and lacks details needed to reproduce the results.

Reply to the Reviewers' comments

Thank you for coordinating the review process for our manuscript, and we sincerely appreciate all the reviewers for their positive evaluations and valuable comments. The reviewer's comments have been carefully considered, and additional experiments have been carried out. The manuscript has been carefully revised to address the reviewers' concerns. The major revisions are marked **in red** in the revised manuscript. The point-to-point responses to the comments are listed below.

Reviewer #2:

In the submitted manuscript Zehang Cui et. al. fabricated Janus membranes using femtosecond laser microfabrication on pre-stretched silicone membrane surface. Specifically, hexagonal grooves were inscribed with help of laser on stretched silicone surface with pores drilled inside the grooves. The top hydrophilic surface was created by reagent treatment whereas inherent hydrophobicity of the silicone at the bottom was utilized to generate Janus membranes. The mechanical stability and performance of the membrane is evaluated under extreme harsh conditions and no loss in performance was observed. Although the substrates are well-characterized and results adequately supported by experimental studies, due to (i) lack of basic correlation between material studied and application (ii) lack of discussion in support of observation and (iii) overall novelty of work, I do not support the publication of this manuscript in a reputed journal like Nature Communication with high impact factor. Please see below my concerns:

1. The details about Mesobiosys, a reagent used for hydrophilization are missing. Basically, authors did not mention how this makes the surface hydrophilic? What is the chemistry driving the interaction between silicone substrate and mesobiosys? Failing to provide the details limits the complete understanding of chemistry driving the substrate performance.

Response: We thank the reviewer for this insightful comment. The chemical composition of the hydrophilic reagent is polyethylene glycol (PEG) solution, which is obtained from the seller (MesoBioSys. Co. Ltd., Wuhan, China). To investigate the interaction between the silicone substrate and the hydrophilic reagent, Fourier transform infrared (FTIR) analysis is performed.

Fig. R1 shows the FTIR spectra of pure silicone, hydrophilic modified silicone and the PEG solution. In the spectra of the silicone, peaks at 3425 and 790 cm^{-1} are assigned to the Si-OH and Si-O-Si, respectively. In the spectra of the hydrophilic reagent, the peaks at 3424, 2870 and 1094 cm^{-1} indicate the -OH, -CH₂O- and C-O-C bonds of PEG, respectively. In the spectra of the hydrophilic modified silicone membrane, the characteristic absorption peaks of PEG at 3424, 2870, 1467, 1368 and 1094 cm^{-1} also appear, which confirms that the PEG is incorporated with the silicone substrate. Compared with PEG solution, the peak at 3424 cm^{-1} , caused by the asymmetric stretching vibration of the -OH functional group, has a slight shift to 3422 cm^{-1} in hydrophilic silicone, which can be attributed to the intermolecular hydrogen bond interactions between the Si-OH of silicone and the terminal hydroxyl group of PEG. Caused by symmetric stretching vibration of the C-O-C functional group, the peak from 800 cm^{-1} to 1500 cm^{-1} of PEG also shifts to smaller wave number and the intensity weakens slightly,

indicates the hydrogen bond interaction established between the Si-OH of silicone and the oxygen atom of C-O-C.

The results show that no obvious new peaks are observed in the spectrum of hydrophilic silicone, indicating that there is physical absorption between the silicone substrate and the hydrophilic reagent, not chemical interactions (Similar interaction mechanism is also discussed in *Sol. Energy Mater. Sol. Cells* **118**, 48-53 (2013), *Appl. Energy*, **86**, 170-174 (2009)).

In conclusion, the Mesobiosys hydrophilic reagent with the main component of PEG polymer, attaches to the silicone membrane via intermolecular hydrogen bond interactions, making the intrinsic hydrophobic silicone surface to be hydrophilic.

Fig. R1 Fourier transform infrared (FTIR) spectra of silicone, hydrophilic modified silicone and PEG solution.

Revision: In accordance with the reviewer’s comments, we have added a discussion of the chemical composition of the hydrophilic coating and the preparation of hydrophilic silicone surfaces to the revised manuscript and supplementary information.

Page 4, Line 99 – Line 102: “*The hydrophilic reagent, composed mainly of hydrophilic polyethylene glycol (PEG), is attached to the pores and grooves (PG channels) via the intermolecular hydrogen bond interactions (Supplementary Fig. 1b). This modification makes the intrinsic hydrophobic PG channels to be hydrophilic (HL, top side),*” has been added.

Additional Figure (*Supplementary Fig. 1b*) and *corresponding caption* have been added to the revised supplementary information.

2. The technique of stretching and creating patterns using femtosecond laser for surface

modification is well known and reported multiple times earlier, for example: <https://doi.org/10.1021/acsami.1c02121>. My concern is stretching silicone surface and patterning it then releasing it to open and close the pores is far from real-life application. The authors are good at patterning the surface but where these stretched Silicone substrates used for fog collection in reality? Much simpler designs are proposed and demonstrated earlier example: <https://doi.org/10.1021/acs.langmuir.2c02697>.

Response: We appreciate the insightful comment from the reviewer. As one of the widely recognized applications of Janus membranes, fog collection has been a feasible solution for addressing water scarcity in arid areas. Indeed, the utilization of femtosecond laser for surface modification is well known in this field. Fog collection has been demonstrated by creating open micropores on aluminium foil (<https://doi.org/10.1021/acsami.1c02121>) and on copper mesh (<https://doi.org/10.1021/acs.langmuir.2c02697>). However, the traditional membranes with open micropores face substantial challenges in harsh environments such as deserts prone to sandstorms. The intense impact of sandstorms poses a significant threat to the longevity of Janus fog collectors, leading to the clogging of open micropores and wear on delicate microstructures and chemical coatings. These issues need to be addressed to ensure the long-term functionality and robustness of fog collectors in such challenging conditions. In fact, water mist occurs only during the early morning hours, so there is no need to leave the Janus membrane open all the time. Our robust Janus membrane leverages its mode-switching capability to selectively open and close microchannels on demand. This innovative design tackles the durability issue by implementing a resilient fog collection method tailored to specific time periods and environmental conditions. As illustrated in Fig. R2, during the early morning, the stretching Janus mode efficiently captures overnight accumulated fog. As the fog disperses and sandstorms approach at noon, the protection mode becomes active, providing resistance against mechanical sandy abrasion and preventing microchannel clogging. Additionally, the protection mode can serve as a long-term storage mode, thereby further enhancing the overall durability of fog collectors. **In conclusion, we believe that the implementation of the mode switching strategy endows the Janus membrane with unique advantages in chemical coating anti-abrasion and microchannel anti-clogging to cope with harsh real-world conditions compared with the existing femtosecond laser-fabricated Janus membrane studies.**

Fig. R2 Schematic illustration of the durable Janus membrane with on-demand mode switching strategy for fog collection in arid deserts. This innovative design tackles the durability issue by implementing a resilient fog collection method tailored to specific time periods and environmental conditions.

Considering the complexity of the practical application environment of the fog collector, the further long-term thermal, humidity and chemical stability performance comparison of membranes in the protected state and open state are also investigated. The temperature of -20°C , 20°C and 60°C , humidity of 30%, 50%, 70% and 90%, chemical immersion into HCl solution ($\text{pH} \sim 4$), NaOH solution ($\text{pH} \sim 10$) and 5 wt.% NaCl solution are adopted to long-term preserve membranes for 10 days. After that, the protected membranes are restretched to the Janus mode at 80% strain, and the weight of collected water (WCW) for 30 minutes is measured to evaluate the stability. Compared with the original state (the freshly prepared Janus membrane) in Fig. 6d, the WCW of these membranes for 30 minutes remains largely unchanged (Fig. R3). **It can be concluded that the temperature, humidity changes and acidic, alkaline, salt corrosion have little effect on the hydrophilic coating.**

Fig. R3 Comparison of the 30-minute WCWs of the Janus membranes in the protected and open states after 10 days of storage at **a** temperature of -20°C , 20°C and 60°C , **b** relative humidity of 30%, 50%, 70% and 90%, **c** chemical immersion of HCl ($\text{pH} \sim 4$), NaOH ($\text{pH} \sim 10$) or 5 wt.% NaCl solution.

Revision: In accordance with the reviewer’s constructive comments, we have added a discussion of the unique advantages of the Janus fog collector in terms of mode switching performance to the revised manuscript. The revisions to the original manuscript are marked **in red**.

Refs. 14 and 15 have been added.

Page 10, Line 283 – Line 291: *“In the real-world harsh condition like the arid desert, the fog collector faces a significant challenge from sandstorm abrasion, during which, the hydrophilic coating is worn and the microchannels are clogged. As an alternative solution, the robust Janus membrane here with mode switching capability can be utilized as a durable fog collector. Fig. 6a illustrates a new operation mechanism adaptable to varying time and environmental conditions. During the early morning, the stretching Janus mode efficiently captures overnight accumulated fog. As the fog disperses and*

sandstorms approach at noon, the protection mode becomes active, providing resistance against mechanical sandy abrasion and preventing microchannel clogging. Additionally, the protection mode can serve as a long-term storage mode, thereby further enhancing the overall durability of fog collectors.” has been added.

Page 11, Line 309 – Line 319: *“Considering the complexity of the practical application environment of the fog collector, the further long-term thermal, humidity and chemical stability performance of the Janus membrane in protection mode are also investigated. In terms of thermal stability, membranes are exposed in the temperature of -20°C , 20°C and 60°C for 10 days considering the temperature difference between day and night in the desert. Compared with the original state (the freshly prepared Janus membrane) in Fig. 6d, the WCW of these membranes for 30 minutes remains largely unchanged, as shown in Fig. 6e. In addition, membranes are kept in the conditions with different relative humidity (30%, 50%, 70% and 90%) and immersed into different chemical solutions (HCl, NaOH and NaCl). As shown in Fig. 6f-g, after 10 days’ treatment, the WCW of a period 30 minutes also keep consistent compared with the original state. The results indicate the outstanding thermal, humidity and chemical stability of the Janus membrane.”* has been added.

Additional Figures (*Fig. 6a, Fig. 6e-g*), *corresponding captions* have been added.

3. Mechanical abrasions are studied here and it’s good to preserve the microstructure and pores but this may not be necessary/needed for a fog collector. Other application like smart clothing maybe more relevant.

Response: We appreciate the reviewer’s insightful comment. Typically, a Janus membrane-based fog collector comprises two critical components: vertically oriented micropores and opposite surface wettability. The opposite wettability is achieved by incorporating hydrophilic or hydrophobic layer on intrinsically hydrophobic or hydrophilic substrates, respectively. In harsh working conditions such as arid deserts, prolonged sandstorm scouring may compromise the unidirectional water penetration of traditional open-pore Janus membranes. This occurs by clogging the micropores, scratching the fragile microstructure and deteriorating the hydrophilic chemical coatings, posing mechanical abrasion as a significant challenge to the durability of Janus fog collectors.

We express our sincere appreciation to the reviewer for suggesting the application in smart clothing. In this regard, our Janus membrane functions as a smart wearable skin patch for simultaneous human motion sensing and sweat collection, as depicted in Fig. R4a. Specifically, a hydrophilic carbon nanotube (MWCNT) coating is employed to construct flexible and electronic robust Janus membranes. During the repeated knee bending, the Janus membrane switches between the two modes, with the hydrophilic conductive MWCNT coating monitoring body motion by varying resistance, and the unidirectional water penetration of the Janus membrane enabling the sweat collection for further biochemical analysis. With the aid of the mode switching strategy, the MWCNTs inside the microgrooves remain preserved during mechanical abrasion (Fig. R4b). To investigate the influence of sweating on the resistance-strain relationship, a wet membrane is prepared by adding 15 μL of 0.9 wt.% NaCl solution. The total resistance of the membrane varies with biaxial stretching strain (Fig.

R4c). The influence of mechanical abrasion on the resistance-strain relationship for both dry and wet Janus membranes is quantified (Fig. R4d-e). The total resistance increases monotonically with strain from $\varepsilon = 0\%$ to $\varepsilon = 80\%$ before and after abrasion, and the resistance increases faster after abrasion because the MWCNTs on the top surface are scratched away. Notably, the wet membrane exhibits a more larger resistance variation compared to the dry membrane because the absorbed water molecules 1) on the hydrophilic MWCNT act as electron donors to increase the resistance and 2) fill the space between nanotubes and forming layers that are less conductive, hindering the electron transfer and resulting in the increase of resistance (*Chem. Eng. J.* **441**, 136103 (2022), *Langmuir* **35**, 4834-4842 (2019)). The experimental optical photographs of the wearable patch and sweat collection before and after abrasion are shown in Fig. R4f, demonstrating the preservation of the sweat collection function. Fig. R4g-h present the smart patch before and after abrasion for the real-time detection of knee bending movement and sweating. Compared with normal walking process, after sweating the baseline and peaks of the resistance signal both increase owing to the introduction of sweat. When sweat evaporation occurs, the resistance baseline and peaks gradually return to the level similar to the dry state. Different motions, such as running and stepping, can be recognized based on the frequency and amplitude of the signal peaks. For fast-moving individuals, signal peaks emerge as small high-frequency peaks. In contrast, the stepping motion manifests as a high peak and a low frequency. Compared with the membrane before abrasion, the motion monitoring and sweat collection functionality maintain intact even after experiencing abrasion. And the signal peaks after abrasion for the same action became even larger because the MWCNT on the top surface is scratched away (Fig. R4d-e). The robust performance against abrasion in body motion sensing and sweat collection highlights the effectiveness of the mode-switching strategy. The integrated functionality allows for both motion monitoring and biochemical data acquisition, making the Janus membrane a promising tool in wearable technology for health monitoring.

Fig. R4 **a** Schematic of the Janus membrane used as a wearable smart patch on human knees for motion sensing and sweat collection. **b** Schematic illustration of hydrophilic MWCNT coating-based mode switching of this conductive Janus membrane. **c** Schematic diagram of the total resistance quantification at different stretching strain values. **d-e** Variation in the total resistance of **d** the dry and **e** the wet Janus membranes at different stretching strains after different abrasion cycles. **f** Optical photograph of the smart patch and close-ups of sweat collection before and after abrasion. **g-h** Resistance sensing signals of the smart patch **g** before and **h** after abrasion for walking, sweating and evaporating and for running and stepping movements monitoring during repeated knee bends.

Revision: In accordance with the reviewer’s suggestions, we have added a discussion on the necessity of mechanical abrasion resistance for a fog collector to the revised manuscript. The revisions to the original manuscript are marked in red.

Page 10, Line 283 – Line 286: “*In the real-world harsh condition like the arid desert, the fog collector faces a significant challenge from sandstorm abrasion, during which, the hydrophilic coating is worn and the microchannels are clogged. As an alternative solution, the robust Janus membrane here with mode switching capability can be utilized as a durable fog collector.*” has been added.

Additional Figure (*Fig. 6a*) and *corresponding captions* have been added.

4. Some redundant tests are carried out after which the performance did not change, to me this

is not a surprise. For example (1) release of strain reduced the pore size from 35 μm to zero. This closure state is called “protective mode” and this phenomenon is as expected. (2) No water flow is observed in closed state and under applied strain state, again quite expected. (3) Silicone membranes are elastomer in nature, and I did not see any novelty that they can be stretched for 20 cycles without loss in performance.

Response: We thank the reviewer for these comments. The variations in pore size, pore spacing and groove width with applied strain are quantified to demonstrate the adjustable nature of the morphology. However, we apologize if there was any confusion caused. In reality, complete closure of micro-pores and prevention of liquid droplet transportation only occur when a specific **threshold of applied pre-strain during laser processing** is exceeded. As shown in Fig. R5, no obvious water penetration is observed on the top side in the released state (termed as “closed state”) with a prestretching strain of 80% during laser processing. In contrast, water penetration occurs in the released state when the prestretching strain is less than 80% (40%, 60% as an example) during laser processing, indicating that the Janus mode at 80% pre-strain possesses a tighter closing state, namely, better mechanical impact resistance, defined as the protection mode.

Fig. R5 Top-view and side-view photographs of PG channels and corresponding unidirectional penetration behaviours of closed-state membranes with prestretching strains of 40%, 60% and 80%. Scale bar: 200 μm .

Indeed, silicone membranes are elastomer and can be stretched for 20 cycles without loss in performance. The experiment of continuous switching between the Janus mode and the protection mode over 20 cycles is performed to investigate not only the elasticity of the membrane but also the durability of the hydrophilic coating on the surface. After a single modification of the membrane with the hydrophilic reagent, the water unidirectional penetration in the stretched state is not observed after only 7 stretching/releasing cycles (Fig. R6). In comparison, repeated hydrophilic modifications are employed to enhance the durability. Specifically, the reagent is applied to the stretched surface, and the membrane is released and restretched for further modification. The process is repeated 3 times to obtain the Janus membrane with robust stretching/releasing switchability over 20 cycles without loss in performance, the result is shown in the manuscript (Fig. 3g).

Fig. R6 Water unidirectional penetration of a Janus membrane with a single-time hydrophilic modification during repeated stretching and releasing cycles. This indicates that the single-time hydrophilic modification is insufficient to maintain the desired unidirectional water transport properties of the membrane throughout multiple cycles of stretching and releasing.

Revision: In accordance with the reviewer’s comments, we have added a discussion on the release of the membrane in a closed state and the influence of repeated stretching on the function of the Janus membrane to the revised manuscript and supplementary information. The revisions to the original manuscript are marked in red.

Page 5, Line 147 – Line 156: *“Considering that cracks or defects may arise in microgrooves during the stretching process, potentially compromising the water unidirectional penetration function, we investigate the impact of repeated stretching on the Janus membrane function. Results show that new cracks only appear during the first three stretch cycles (Supplementary Fig. 8). To mitigate the effects of these cracks on Janus membrane performance, a repeated hydrophilic modification approach is employed. Compared with the water unidirectional penetration function of Janus membrane modified one time in stretched state (renders ineffective after just 7 stretching/releasing cycles, Supplementary Fig. 9), the robust switchability observed over 20 cycles indicates that the repeated stretching during the modification process eliminates the impact of microgrooves surface cracks on Janus membrane performance.”* has been added.

Additional Figures (*Supplementary Fig. 8, Supplementary Fig. 9*), *corresponding captions* have been added in the revised supplementary information.

5. Change in penetration time after stretching cycles (Fig 6 in SI), how to authors support this observation? Merely stating “consumption of hydrophilic coating” does not suffice. Can authors quantify the results in term of consumed reagent? Or was this observation due to change in the intrinsic material properties of silicone like cracks/defect generated? Was hydrophilicity generated on the reagent used bound to silicone covalently/ionically/supramolecularly/van der

waals attraction?

Response: We appreciate the reviewer's insightful question. As illustrated in Supplementary Fig. 6, the increase in penetration time with repeated cycles is mainly due to the consumption of the hydrophilic coating by the successively added droplets. To support this observation, we quantify the consumption of the hydrophilic coating by Fourier transform infrared (FTIR) analysis. The continuous addition of droplets is performed on the silicone substrate with a hydrophilic reagent coating, and the FTIR spectra are recorded for the 0, 5th, 10th, 15th and 20th droplets. As shown in Fig. R7, with the continuous addition of droplets from 0 to the 20th drop, the intensity of the characteristic peaks at 3422 and 2867 cm^{-1} (corresponding to the -OH and -CH₂O of PEG) progressively decreases, **which indicates that the decreasing PEG content on the silicone substrate, namely, the consumption of the hydrophilic coating.**

Fig. R7 **a** Fourier transform infrared (FTIR) spectrum of hydrophilic modified silicone with continuous addition of droplets. **b** High-resolution FTIR spectrum of the region in the dotted box in **a**.

Indeed, cracks appear inside microgrooves during the stretching and releasing process. Specifically, **cracks/defects are generated inside the fabricated microgrooves during the first three stretching cycles; after that, the morphology remains constant, and no new cracks/defects appear.** As shown in Fig. R8, as the number of stretching cycles increases from 0 to 3, the surface roughness Ra increases from ~ 2.72 to $3.51 \mu\text{m}$, which is attributed to the generation of cracks. Moreover, Ra remains constant as the number of stretching cycles increases from 3 to 23, which suggests that no new cracks appear. Notably, repeated hydrophilic modifications are employed in the experiments to avoid the effects of cracks on the Janus membrane performance. The reagent is applied to the stretched surface, after which the membrane is released and restretched for further modification. This process is repeated 3 times to obtain the hydrophilic top surface of the Janus membrane. This means that all the tests are carried out after the 4th stretching cycle. Therefore, the effect of cracks formation on the change of the water unidirectional penetration time is excluded and the only influencing factor is the consumption of the hydrophilic coating.

Fig. R8 SEM images, Confocal Laser Scanning Microscopy (CLSM) images and profile comparison of microgroove morphologies after different stretching and releasing cycles. Scale bar: 50 μm .

The Fourier transform infrared (FTIR) spectrum in Fig. R9 shows that no obvious new chemical bonds are observed on the silicone substrate after hydrophilic modification, which indicates that there is physical absorption between the silicone substrate and the hydrophilic reagent, which is also discussed in the response to Question #1. In conclusion, the Mesobiosys hydrophilic reagent with the main component of PEG polymer, attaches to the silicone membrane via **intermolecular hydrogen bond interactions**, making the intrinsic hydrophobic silicone surface to be hydrophilic.

Fig. R9 Fourier transform infrared (FTIR) spectrum comparison of hydrophilic silicone and PEG solution.

Revision: In accordance with the reviewer's comments, we have added a discussion on the quantification of hydrophilic coating consumption, the influence of repeated stretching on Janus membrane function and the mechanism of hydrophilic modification to the revised manuscript and supplementary information. The revisions to the original manuscript are marked **in red**.

Page 4, Line 99 – Line 102: “*The hydrophilic reagent, composed mainly of hydrophilic polyethylene glycol (PEG), is attached to the pores and grooves (PG channels) via the intermolecular hydrogen bond interactions (Supplementary Fig. 1b). This modification makes the intrinsic hydrophobic PG channels to be hydrophilic (HL, top side),*” has been added.

Page 5, Line 147 – Line 156: “*Considering that cracks or defects may arise in microgrooves during the stretching process, potentially compromising the water unidirectional penetration function, we investigate the impact of repeated stretching on the Janus membrane function. Results show that new cracks only appear during the first three stretch cycles (Supplementary Fig. 8). To mitigate the effects of these cracks on Janus membrane performance, a repeated hydrophilic modification approach is employed. Compared with the water unidirectional penetration function of Janus membrane modified one time in stretched state (renders ineffective after just 7 stretching/releasing cycles, Supplementary Fig. 9), the robust switchability observed over 20 cycles indicates that the repeated stretching during the modification process eliminates the impact of microgrooves surface cracks on Janus membrane performance.*” has been added.

For better understanding, *the SEM images of PG channels before and after durability tests in Fig. 5d* have been replaced with the images after three pre-stretching cycles.

Additional Figures (*Supplementary Fig. 1b, Supplementary Fig. 7, Supplementary Fig. 8, Supplementary Fig. 9*) and *corresponding captions* have been added to the revised supplementary information.

6. In line 136/137 authors declare “In real-world applications, membranes are usually subjected to harsh conditions such as contact abrasion” were are such conditions? Can authors specify such applications with respect to current studies where this “protective mode” can be utilized specifically w.r.t silicone membranes?

Response: We thank the reviewer for this helpful comment. Since silicone has both elasticity and wear resistance, we choose it to prepare wear-resistant devices. One application is Janus fog collection membranes in harsh desert conditions, where the membranes are usually subjected to long-term sandstorm scouring, which can be considered mechanical abrasion. The continuous wear on the microstructures and coatings of traditional open micropore Janus fog collectors pose a significant threat to their service life. Taking advantage of the on-demand mode switching capability to open and close the microchannels, the robust Janus membrane here solves this tough problem via a durable fog collection manner based on time periods and conditions (Fig. R2). The details are discussed in the response to Question #2. The other application is wearable smart patches (Fig. R4). During daily use

and washing, wearable patches are also subject to abrasion. For instance, the membrane is released to protection mode before washing to enhance its durability.

Fig. R2 Schematic illustration of the durable Janus membrane with on-demand mode switching strategy for fog collection in arid deserts.

Fig. R4 **a** Schematic of the Janus membrane used as a wearable smart patch on human knees for motion sensing and sweat collection. **b** Schematic illustration of hydrophilic MWCNT coating-based mode switching of this conductive Janus membrane. **c** Schematic diagram of the total resistance quantification at different stretching strain values. **d-e** Variation in the total resistance of **d** the dry and **e** the wet Janus membranes at different stretching strains after different abrasion cycles. **f** Optical photograph of the smart patch and close-ups of sweat collection before and after abrasion. **g-h** Resistance sensing signals of the smart patch **g** before and **h** after abrasion for walking, sweating and

evaporating and for running and stepping movements monitoring during repeated knee bends.

Revision: In accordance with the reviewer's comments, we have added a discussion of the specific application scenarios of this mode-switching Janus fog collector to the revised manuscript, and the revisions to the original manuscript are marked **in red**.

Page 10, Line 286 – Line 291: “*Fig. 6a illustrates a new operation mechanism adaptable to varying time and environmental conditions. During the early morning, the stretching Janus mode efficiently captures overnight accumulated fog. As the fog disperses and sandstorms approach at noon, the protection mode becomes active, providing resistance against mechanical sandy abrasion and preventing microchannel clogging. Additionally, the protection mode can serve as a long-term storage mode, thereby further enhancing the overall durability of fog collectors.*” has been added.

Page 6, Line 158 – Line 159: To avoid confusion, “*In real-world applications, membranes are usually subjected to harsh conditions such as contact abrasion.*” has been deleted.

Additional Figure (*Fig. 6a*) and *corresponding caption* have been added.

7. During the 2000 cycle abrasion test, the groove depth decreased from 82 μm to 43 μm and no significant performance change was observed (unidirectional water flow behavior). But this is merely because the test was designed in a manner such that complete erosion of grooves does not take place. What is missing is insight on water transport properties, was there a significant change in resistance to water flow?

Response: We thank the reviewer for this helpful comment. The unidirectional water penetration from the bottom to the top side of the Janus membrane can be divided into three stages. Water is first pumped into the micropores and wet them (Stage I). Subsequently, water flows and fills all the microgrooves, forming a thin film on the hydrophilic side (Stage II). Finally, a convex droplet emerges rapidly and spreads on the top side (stage III) (Fig. R10). The magnitude of the driving force and resistance in Stage I remain constant before and after abrasion because the micropore is protected.

Fig. R10 *In-situ* observation and schematic of water unidirectional penetration and spreading on the top hydrophilic PG channels side. Scale bar: 200 μm .

During the 2000 cycle abrasion test, the groove depth decreases from 82 μm to 43 μm , and the water unidirectional penetration time increases from 3.8 s to 6.5 s, as shown in Fig. 4e. Here the water unidirectional transport mechanism before and after abrasion is thoroughly investigated. As water flows inside the hydrophilic gridded microgrooves during Stage II, the main **driving force** is derived from the capillary force $F_c \approx 2\gamma h \cos\theta$ (Langmuir **14**, 3937-3943 (1998), *Nat. Mater.* **17**, 935-942 (2018)), where γ is the surface tension of water, h is the groove depth and θ is the water contact angle. The **resistance** is derived from the viscous force $F_v = 3\eta x u / \varepsilon \zeta(\varepsilon)$ (Langmuir **35**, 8131-8143 (2019), *Microfluid. Nanofluid.* **15**, 309-326 (2013)), where η is the water viscosity, x is the water transport distance inside the microgroove, u is the flow velocity, and the aspect ratio of the microgrooves is $\varepsilon = h/w$ (w is the groove width). Additionally, $\zeta^{-1}(\varepsilon) \approx 1 + 0.671004\varepsilon + 4.169711\varepsilon^2$. Theoretically, after the 2000 cycle abrasion test, the groove depth decreases from 82 μm to 43 μm , the estimated driving force F_c decreases from 1.17×10^{-5} N to 6.16×10^{-6} N, and the estimated resistance F_v decreases from 1.09×10^{-7} N to 4.91×10^{-8} N, thus the water penetration time increases (Fig. R11).

In summary, with increasing abrasion cycle, in stage II, the driving forces F_c decreases while the resistance F_v increases. Notably, the magnitude of F_v is much lower than that of F_c . Therefore, $F = F_c - F_v \approx F_c$, and water flow is mainly driven by capillary forces.

Fig. R11 Dominant driving force analysis of water flow in microgrooves with increasing abrasion cycles.

Additionally, complete abrasion of microgrooves after approximately 6000 abrasion cycles is conducted to validate this theory. As shown in Fig. R12, the droplet cannot penetrate from the bottom to the top side when the microgrooves are fully abraded. The driving force in Stage II approaches zero as the microgroove depth h diminishes to 0. The intrinsic hydrophobicity of silicone is exposed when the microgrooves are scratched away.

Fig. R12 a Optical and CLSM images of PG channels after 6000 abrasion cycles at stretching strains of 0% and 80%. Scale bars: 200 μm . **b** Optical image and schematic of a water droplet that cannot penetrate from the bottom to the top side.

Revision: In accordance with the reviewer’s comments, we have added a more detailed discussion of the water flow behavior inside the microgrooves during Stage II to the revised manuscript and supplementary information (*Supplementary Note 1*). The revisions to the original manuscript are marked in red.

Page 7, Line 198 – Line 199: “*The gravity of the droplet that has not yet entered the micropore F_g acts a resistance in the vertical advancing direction.*” has been rearranged.

Page 7, Line 203 – Line 226: “*the surface tension of the water (F_a) and the capillary force within the laser drilling asymmetric conical micropores (F_b), collectively drive the droplet from the hydrophobic bottom side into the pore microchannels⁴. And the driving forces can be described as follows:*

$$F_w = \gamma l (\cos \theta_a - \cos \theta_r) \quad (1)$$

$$F_a = \frac{\pi \gamma d^2}{2r_1} \quad (2)$$

$$F_b = 2\pi \gamma r_2 \cos \left| \theta + \frac{\alpha}{2} \right| \quad (3)$$

where γ is the surface tension of water, l is the effective contact length of the droplets, θ_a and θ_r are the local advancing and receding contact angles, d is diameter of the micropore, r_1 is the radius of the water droplet, r_2 is the radius of the three-phase contact line inside the micropore, θ is the water

contact angle, and α is the taper angle of micropore, respectively. And the magnitude of driving force F_w , F_a and F_b can be supposed to remain constant before and after abrasion because of micropores preservation in protection mode.” has been rearranged.

Page 8, Line 232 – Line 240: “*In this equation:*

$$F_v = \frac{3\eta xu}{\varepsilon \zeta(\varepsilon)} \quad (6)$$

where h is the groove depth, η is the water viscosity, x is the water transport distance inside the microgroove, u is the flow velocity, and the aspect ratio of the microgrooves is $\varepsilon = h/w$ (w is the groove width).” has been rearranged.

Page 9, Line 248 – Line 250: “*When complete erosion of the microgrooves occurs after about 6000 abrasion cycles, namely $h \sim 0$, water droplets cannot penetrate from the bottom to the top side (Supplementary Fig. 14).*” has been added.

Additional Figure (*Supplementary Fig. 14*) and *corresponding caption* have been added in the revised supplementary information.

8. What are the properties of Janus polymer used for comparative studies (Supplementary Fig. 9). Without providing the membrane characteristic like thickness, pore size, contact angle at hydrophilic and hydrophobic surface etc. such tests will be inaccurate/incomplete.

Response: We thank the reviewer for this helpful comment. To compare the hydrophilicity and durability when exposed to air, a polymer-based Janus membrane with only micropores is fabricated as a control sample. As shown in Fig. R13, the thickness, micropore size and pore spacing of the control open micropore Janus membrane is 500 μm , 40 μm and 150 μm , respectively, same with the PG channels membranes in stretched state. Single-sided hydrophilic reagent treatment is also used to modify the top side of the control membrane to be hydrophilic with a CA of $\sim 15^\circ$, and the bottom side exhibits intrinsic silicone hydrophobicity with a CA of $\sim 120^\circ$.

Fig. R13 Optical microscopy images of the hydrophilic top and hydrophobic bottom sides of the traditional open micropore Janus membrane and the corresponding water contact angles. Scale bar: 100 μm .

Revision: In accordance with the reviewer’s comments, we have added additional characteristics of the comparative open-pore Janus membrane to the supplementary information.

Additional Figure (*Supplementary Fig. 12b*) and *corresponding caption* have been added in the revised supplementary information.

9. The magnitude of driving force is expected to remain constant before and after abrasion because the hydrophilic surface was protected during abrasion and the active thickness available for water flow is almost constant.

Response: We thank the reviewer for this helpful comment. As discussed in the response to question #7, the active thickness available for water flow decreases after abrasion, resulting in a reduced driving force. The unidirectional water penetration from the bottom to the top side of the Janus membrane can be divided into three stages. Water is first pumped into the micropores and wet them (Stage I). Subsequently, water flows and fills all the microgrooves, forming a thin film on the hydrophilic side (Stage II). Finally, a convex droplet emerges rapidly and spreads on the top side (Stage III) (Fig. R10). The magnitude of the driving force in Stage I should remain constant before and after abrasion because of preservation of the micropore during protection.

Fig. R10 *In situ* observation and corresponding schematic of water unidirectional penetration and spreading on the top hydrophilic PG channels side. Scale bar: 200 μm .

When water flows in the gridded microgrooves, the driving force can be described as follows:

$$F = F_c - F_v \quad (\text{R1})$$

where F_c is the capillary force and F_v is the viscous force. In this equation (*J. Phys. Chem. C* **115**, 18761-18769 (2011), *Langmuir* **14**, 3937-3943 (1998), *Nat. Mater.* **17**, 935-942 (2018)):

$$F_c = - \frac{dE}{dx} = \gamma[(\cos \theta - 1)w + 2h \cos \theta] \quad (\text{R2})$$

where E , x , γ and θ are the free energy change, water flow filled groove length, water surface tension and water contact angle, respectively. For water flow inside hydrophilic microgrooves (with depth h and width w), $\cos \theta \approx 1$, that is:

$$F_c \approx 2\gamma h \cos \theta \quad (\text{R3})$$

The values of F_{c1} and F_{c2} (before and after 2000 abrasion cycles) are estimated to be 1.17×10^{-5} N and 6.16×10^{-6} N, respectively. For the viscous force (Langmuir **35**, 8131-8143 (2019), *Microfluid. Nanofluid.* **15**, 309-326 (2013)):

$$F_v = \frac{3\eta xu}{\varepsilon\zeta(\varepsilon)} \quad (\text{R4})$$

where η is the water viscosity, u is the flow velocity, and the aspect ratio of the microgrooves is $\varepsilon = h/w$. Additionally, for $\zeta^{-1}(\varepsilon) \approx 1+0.671004\varepsilon + 4.169711\varepsilon^2$, the ε values before and after 2000 abrasion cycles are established as 1.41 and 0.74, and the values of F_{v1} and F_{v2} (before and after 2000 abrasion cycles) are calculated to be 1.09×10^{-7} N and 4.91×10^{-8} N, respectively. We can conclude that in the microgrooves, capillary force is the main driving force, the magnitude of which is determined by h .

In conclusion, both the magnitude of the driving force and the active thickness available for water flow decrease with the increasing abrasion cycles.

Revision: In accordance with the reviewer's comments, we have added a detailed discussion of the dominant driving force analysis of water flow in microgrooves with increasing abrasion cycles to the revised manuscript and supplementary information (*Supplementary Note 1*). The revisions to the original manuscript are marked in red.

Page 7, Line 198 – Line 199: “*The gravity of the droplet that has not yet entered the micropore F_g acts a resistance in the vertical advancing direction.*” has been rearranged.

Page 7, Line 203 – Line 226: “*the surface tension of the water (F_a) and the capillary force within the laser drilling asymmetric conical micropores (F_b), collectively drive the droplet from the hydrophobic bottom side into the pore microchannels⁴. And the driving forces can be described as follows:*

$$F_w = \gamma l (\cos \theta_a - \cos \theta_r) \quad (1)$$

$$F_a = \frac{\pi\gamma d^2}{2r_1} \quad (2)$$

$$F_b = 2\pi\gamma r_2 \cos \left| \theta + \frac{\alpha}{2} \right| \quad (3)$$

where γ is the surface tension of water, l is the effective contact length of the droplets, θ_a and θ_r are the local advancing and receding contact angles, d is diameter of the micropore, r_1 is the radius of the water droplet, r_2 is the radius of the three-phase contact line inside the micropore, θ is the water contact angle, and α is the taper angle of micropore, respectively. And the magnitude of driving force F_w , F_a and F_b can be supposed to remain constant before and after abrasion because of micropores preservation in protection mode.” has been rearranged.

Page 8, Line 232 – Line 240: “*In this equation:*

$$F_v = \frac{3\eta xu}{\varepsilon\zeta(\varepsilon)} \quad (6)$$

where h is the groove depth, η is the water viscosity, x is the water transport distance inside the microgroove, u is the flow velocity, and the aspect ratio of the microgrooves is $\varepsilon = h/w$ (w is the groove

width.” has been rearranged.

Page 9, Line 248 – Line 250: “*When complete erosion of the microgrooves occurs after about 6000 abrasion cycles, namely $h \sim 0$, water droplets cannot penetrate from the bottom to the top side (Supplementary Fig. 14).*” has been added.

Additional Figure (*Supplementary Fig. 14*) and *corresponding caption* have been added in the revised supplementary information.

10. During rigorous durability tests and fog-collection application test, efforts are dedicated to mechanical stability. How about the performance under different relative humidity conditions for long-term? Can here, the tests should be performed under two conditions (1) protected morphology and (2) open pore morphology? This will inform about the overall stability the designed membrane combined with the coatings and physical attributes.

Response: We thank the reviewer for these helpful suggestions. To further investigate the robustness of this Janus membrane, additional long-term humidity stability tests are preformed, as shown in Fig. R14.

Four homemade chambers with relative humidity of 30%, 50%, 70% and 90% are adopted for long-term preservation test. Three membranes in the protected state and three membranes in the open state are placed in each homemade chamber and stored for 10 days. After that, the protected membranes are restretched to the Janus mode at 80% strain, and the weight of collected water (WCW) after 30 minutes’ fog collecting is recorded to evaluate the stability of the membranes, as shown in Fig. R14. The results show that humidity does not affect the performance of the protected membranes and open pore membranes, allowing the membranes to remain almost the same fog collecting capability under different humidity conditions compared with the original state in the manuscript (as shown in Fig. 6d).

Fig. R14 Comparison of the 30-minute WCWs of the Janus membranes in protected and open states after 10 days of storage at relative humidity of 30%, 50%, 70% and 90%, respectively.

Revision: In accordance with the reviewer’s comments, we have added a discussion on the long-term

humidity stability to the revised manuscript. The revisions to the original manuscript are marked in red.

Page 11, Line 309 – Line 311: “*Considering the complexity of the practical application environment of the fog collector, the further long-term thermal, humidity and chemical stability performance of the Janus membrane in protection mode are also investigated.*” has been added.

Page 11, Line 315 – Line 319: “*In addition, membranes are kept in the conditions with different relative humidity (30%, 50%, 70% and 90%) and immersed into different chemical solutions (HCl, NaOH and NaCl). As shown in Fig. 6f-g, after 10 days’ treatment, the WCW of a period 30 minutes also keep consistent compared with the original state. The results indicate the outstanding thermal, humidity and chemical stability of the Janus membrane.*” has been added.

Additional Figure (Fig. 6f) and *corresponding caption* have been added in the revised supplementary information.

11. Study summarized in figure 12 show importance of the protective mode, but this may be impractical in realistic environment.

Response: We thank the reviewer for this comment. Taking the fog collector as an example, the mechanical durability must be considered in real-world environments. **In extreme environments such as deserts, sandstorming abrasion severely threatens the service life of the fog collectors, possibly by clogging microchannels, scratching away microstructures and chemical coatings with no protection.** Considering these realistic challenges, as shown in **Fig. R2** in the response to Question #2, the proposed mode switching strategy here is an alternative solution to enhance the practical durability of Janus membranes. The other example is the wearable devices, where the durability against mechanical abrasion is also important. The detailed discussion is given in the response to Questions #2 and #3.

Revision: In accordance with the reviewer’s constructive comments, we have added a discussion on the importance and practicality of the protection mode of this Janus membrane to the revised manuscript. The revisions to the original manuscript are marked in red.

Page 10, Line 283 – Line 291: “*In the real-world harsh condition like the arid desert, the fog collector faces a significant challenge from sandstorm abrasion, during which, the hydrophilic coating is worn and the microchannels are clogged. As an alternative solution, the robust Janus membrane here with mode switching capability can be utilized as a durable fog collector. Fig. 6a illustrates a new operation mechanism adaptable to varying time and environmental conditions. During the early morning, the stretching Janus mode efficiently captures overnight accumulated fog. As the fog disperses and sandstorms approach at noon, the protection mode becomes active, providing resistance against mechanical sandy abrasion and preventing microchannel clogging. Additionally, the protection mode can serve as a long-term storage mode, thereby further enhancing the overall durability of fog*”

collectors.” has been added.

12. Sandstrom is a specific case that supports this study, other environmental factors like change in temperature, humidity (mentioned above) are not studied which makes it difficult to visualize the applicability of this material under realistic conditions.

Response: We thank the reviewer for these helpful suggestions. In addition to the mechanical stability and humidity stability discussed in the response to Question #10, we also investigate the thermal stability, as shown in Fig. R15.

Taking the application of fog collector as an example, a long-term preservation test is conducted to assess the performance of the samples under different temperature conditions. Considering the temperature difference between day and night in desert climates, temperatures of -20°C (refrigerator), 20°C (room condition) and 60°C (oven) are adopted for sample long-term preservation test. At each temperature condition, three membranes in the protected state and three membranes in the open state are stored for 10 days. After that, the protected membranes are restretched to the Janus mode at 80% strain, and the weight of collected water (WCW) for 30 minutes is measured to evaluate the thermal stability. The results show that the WCW decreases slightly with the increasing temperature, and the fog collection capability of both the protected and open pore membranes does not significantly decrease compared with that of the original state in the manuscript (as shown in Fig. 6d). It can be concluded that temperature changes have little effect on hydrophilic coating, endowing the Janus membrane with thermal stability.

In summary, the prepared Janus membrane possesses outstanding mechanical, humidity and thermal stability to cope with harsh practical conditions.

Fig. R15 Comparison of the 30-minute WCWs of the Janus membranes in the protective and open states after 10 days of exposure at temperatures of -20°C , 20°C and 60°C .

Revision: In accordance with the reviewer’s comments, we have added a discussion on the thermal stability of the Janus membrane to the revised manuscript. The revisions to the original manuscript are marked in red.

Page 11, Line 309 – Line 315: “*Considering the complexity of the practical application environment of the fog collector, the further long-term thermal, humidity and chemical stability performance of the Janus membrane in protection mode are also investigated. In terms of thermal stability, membranes are exposed in the temperature of -20°C , 20°C and 60°C for 10 days considering the temperature difference between day and night in the desert. Compared with the original state (the freshly prepared Janus membrane) in Fig. 6d, the WCW of these membranes for 30 minutes remains largely unchanged, as shown in Fig. 6e.*” has been added.

Additional Figure (*Fig. 6e*) and *corresponding caption* have been added.

Finally, we thank the reviewer again for these thoughtful comments. The manuscript has greatly benefited from these insightful suggestions.

Reviewer #3:

The author has demonstrated the feasibility of a Janus membrane in a multitude of applications ranging from multiphase separation devices to fog harvesting and wearable health-monitoring patches. Overall, this is a decent research topic which can be noteworthy for researchers' studying Janus membrane fabrication and separation technology. However, in this research paper, there are some important queries which should be addressed:

1. Starting with the abstract, kindly highlight the best outcome of the fabricated Janus membrane for better understanding.

Response: We are grateful for the positive comments. We have polished the abstract and introduction section with respect to the best outcome parameters of the robust Janus membrane in the revised manuscript.

Revision:

Page 1, Line 29 – Line 32: “*The protection mode imparts the Janus membrane robustness to reserve water unidirectional penetration under harsh conditions, such as 2000 cycles mechanical abrasion, 10 days exposure in air and other rigorous tests (sandpaper abrasion, finger rubbing, sand impact and tape peeling).*” has been rearranged.

Page 1, Line 33 – Line 34: “*The Janus membrane serves as a fog collector to demonstrate its unwavering mechanical durability in harsh real-world conditions.*” has been added.

Page 3, Line 85 – Line 87: “*In the proof-of-concept application, the Janus membrane serves as a fog collector, demonstrating its mechanical stability in harsh real-world environments.*” has been rearranged.

2. Very importantly, the generation of hydrophilic pores via femtosecond laser must be well in terms of high scientific discussion. The mechanism must be included.

Response: We thank the reviewer for this insightful suggestion. A discussion of the generation of hydrophilic microgrooves and pores has been added. Herein, femtosecond laser processing and surface hydrophilic modification techniques are adopted. Specifically, femtosecond laser microfabrication is employed to produce microgrooves and micropores, and the hydrophilicity is obtained by hydrophilic reagent modification. The MesoBioSys hydrophilic reagent Mesophilic-2000 (hydrophilic polymer solution that contains PEG) is used for hydrophilic modification of the sample surface. The hydrophilic modification mechanism is investigated and validated in Fig. R16.

Fig. R16 shows the FTIR spectra of pure silicone, hydrophilic modified silicone and the PEG solution. In the spectra of the silicone, peaks at 3425 and 790 cm^{-1} are assigned to the Si-OH and Si-O-Si, respectively. In the spectra of the hydrophilic reagent, the peaks at 3424, 2870 and 1094 cm^{-1} indicate the -OH, -CH₂O- and C-O-C bonds of PEG, respectively. In the spectra of the hydrophilic modified

silicone membrane, the characteristic absorption peaks of PEG at 3424, 2870, 1467, 1368 and 1094 cm^{-1} also appear, which confirms that the PEG is incorporated with the silicone substrate. Compared with PEG solution, the peak at 3424 cm^{-1} , caused by the asymmetric stretching vibration of the -OH functional group, has a slight shift to 3422 cm^{-1} in hydrophilic silicone, which can be attributed to the intermolecular hydrogen bond interactions between the Si-OH of silicone and the terminal hydroxyl group of PEG. Caused by symmetric stretching vibration of the C-O-C functional group, the peak from 800 cm^{-1} to 1500 cm^{-1} of PEG also shifts to smaller wave number and the intensity weakens slightly, indicates the hydrogen bond interaction established between the Si-OH of silicone and the oxygen atom of C-O-C.

The results show that no obvious new peaks are observed in the spectrum of hydrophilic silicone, indicating that there is physical absorption between the silicone substrate and the hydrophilic reagent, not chemical interactions (Similar interaction mechanism is also discussed in *Sol. Energy Mater. Sol. Cells* **118**, 48-53 (2013), *Appl. Energy*, **86**, 170-174 (2009)).

Fig. R16 Fourier transform infrared (FTIR) spectra of silicone, hydrophilic silicone and PEG solution.

Revision: In accordance with the reviewer’s suggestion, we have added a discussion of the hydrophilic modification mechanism to the revised manuscript and supplementary information.

Page 4, Line 99 – Line 102: “*The hydrophilic reagent, composed mainly of hydrophilic polyethylene glycol (PEG), is attached to the pores and grooves (PG channels) via the intermolecular hydrogen bond interactions (Supplementary Fig. 1b). This modification makes the intrinsic hydrophobic PG channels to be hydrophilic (HL, top side),*” has been added.

Additional Figure (*Supplementary Fig. 1b*) and *corresponding captions* have been added to the revised supplementary information.

3. The objective of this research study is in premature stage and must be well elaborated (especially last paragraph).

Response: We thank the reviewer for this helpful comment. Based on the excellent mechanical, thermal, humidity and chemical stability of this robust Janus membrane, we have elaborated upon the research objective of this article and added additional discussion about the application prospects of durable fog collectors in the revised manuscript.

Revision:

Page 10, Line 283 – Line 291: *“In the real-world harsh condition like the arid desert, the fog collector faces a significant challenge from sandstorm abrasion, during which, the hydrophilic coating is worn and the microchannels are clogged. As an alternative solution, the robust Janus membrane here with mode switching capability can be utilized as a durable fog collector. Fig. 6a illustrates a new operation mechanism adaptable to varying time and environmental conditions. During the early morning, the stretching Janus mode efficiently captures overnight accumulated fog. As the fog disperses and sandstorms approach at noon, the protection mode becomes active, providing resistance against mechanical sandy abrasion and preventing microchannel clogging. Additionally, the protection mode can serve as a long-term storage mode, thereby further enhancing the overall durability of fog collectors.”* has been added.

Page 11, Line 309 – Line 325: *“Considering the complexity of the practical application environment of the fog collector, the further long-term thermal, humidity and chemical stability performance of the Janus membrane in protection mode are also investigated. In terms of thermal stability, membranes are exposed in the temperature of -20°C , 20°C and 60°C for 10 days considering the temperature difference between day and night in the desert. Compared with the original state (the freshly prepared Janus membrane) in Fig. 6d, the WCW of these membranes for 30 minutes remains largely unchanged, as shown in Fig. 6e. In addition, membranes are kept in the conditions with different relative humidity (30%, 50%, 70% and 90%) and immersed into different chemical solutions (HCl, NaOH and NaCl). As shown in Fig. 6f-g, after 10 days’ treatment, the WCW of a period 30 minutes also keep consistent compared with the original state. The results indicate the outstanding thermal, humidity and chemical stability of the Janus membrane. What is worth noting that the core novelty of this work lies in the proposal of the mode switching strategy that can significantly enhance the mechanical durability of the Janus membrane regardless of the specific type or composition of the hydrophilic coating. Here, an alternative PVA/silica hydrophilic coating is introduced for HCl and NaOH solution immersion tests. We can expect that this not only demonstrates the universal hydrophilic coating protection effect of the mode switching strategy, but also broadens the range of options for hydrophilic coatings that are both mechanically and chemically durable.”* has been added.

Additional Figures (Fig. 6a, Fig. 6e-g) and corresponding captions have been added.

Fig. 6 a Schematic illustration of the durable Janus membrane with on-demand mode switching strategy for fog collection in arid deserts.

Fig. 6 e The WCW changes in 30 minutes of the Janus membranes after 10 days of exposure at a temperature of -20°C , 20°C and 60°C . **f** The WCW changes in 30 minutes of the Janus membranes after 10 days of exposure at a relative humidity of 30%, 50%, 70% and 90%. **g** The WCW changes in 30 minutes of the Janus membranes after 10 days immersion in HCl solution ($\text{pH} \sim 4$), NaOH solution ($\text{pH} \sim 10$) and 5 wt.% NaCl solution.

4. What can we gain from Figure 4 (e)? Please come up with valid references for better readability. The increased penetration time is almost twice as the original time. Thus, the mechanical stability seems to be bit uncertain.

Response: We thank the reviewer for this insightful suggestion. Fig. 4e is adopted to characterize the abrasion depth of membranes and quantify the influence of mechanical abrasion on water unidirectional penetration performance, the membrane samples subjected to different abrasion cycles (0, 500, 1000, 1500 and 2000 cycles) are measured. The abrasion depth is defined by the profile height difference between the worn area and original surface. To enhance the readability, Fig. 4e has been rearranged and the traditional Janus membrane with fixed open grooves and pores is employed as a control sample to compare the mechanical durability with our robust Janus membrane with mode switching capability (Fig. R17). With the increasing abrasion cycles to 2000, the abrasion depth increases from zero to $42 \mu\text{m}$.

Fig. R17 The influence of liner abrasion cycles on abrasion depth and water unidirectional penetration time of Janus membranes in the protective and open states.

The unidirectional water penetration from the bottom to the top side of the Janus membrane can be divided into three stages. Water is first pumped into the micropores and wet them (Stage I). Subsequently, water flows and fills all the microgrooves, forming a thin film on the hydrophilic side (Stage II). Finally, a convex droplet emerges rapidly and spreads on the top side (stage III) (Fig. R18). The magnitude of the driving force and resistance in Stage I remain constant before and after abrasion because the micropore is protected.

Fig. R18 *In-situ* observation and schematic of water unidirectional penetration and spreading on the top hydrophilic PG channels side. Scale bar: 200 μm.

As water flows inside the hydrophilic gridded microgrooves during Stage II, the main **driving force** is derived from the capillary force $F_c \approx 2\gamma h \cos\theta$ (Langmuir **14**, 3937-3943 (1998), *Nat. Mater.* **17**, 935-942 (2018)), where γ is the surface tension of water, h is the groove depth and θ is the water contact

angle. The **resistance** is derived from the viscous force $F_v = 3\eta xu/\varepsilon\zeta(\varepsilon)$ (Langmuir 35, 8131-8143 (2019), *Microfluid. Nanofluid.* 15, 309-326 (2013)), where η is the water viscosity, x is the water transport distance inside the microgroove, u is the flow velocity, and the aspect ratio of the microgrooves is $\varepsilon = h/w$ (w is the groove width). Additionally, $\zeta^{-1}(\varepsilon) \approx 1 + 0.671004\varepsilon + 4.169711\varepsilon^2$. Theoretically, after the 2000 cycle abrasion test, the groove depth decreases from 82 μm to 43 μm , the estimated driving force F_c decreases from 1.17×10^{-5} N to 6.16×10^{-6} N, and the estimated resistance F_v decreases from 1.09×10^{-7} N to 4.91×10^{-8} N. We can conclude that in the microgrooves, capillary force is the main driving force, the magnitude of which is determined by h (Fig. R19). Thus, the water penetration time increases from 3.8 s to 6.5 s (the function is well preserved) because of the inevitable abrasion of silicone. However, compared to the traditional Janus membrane with fixed open grooves and pores, the unidirectional penetration of water fails after only 300 abrasion cycles.

Fig. R19 Dominant driving force analysis of water flow in microgrooves with increasing abrasion cycles.

Additionally, complete abrasion of microgrooves after approximately 6000 abrasion cycles is conducted. As shown in Fig. R20, the droplet cannot penetrate from the bottom to the top side when the microgrooves are fully abraded. **The results indicate that, compared to the traditional Janus membrane with fixed open grooves and pores, the mechanical durability has been improved by at least 20 times.**

Fig. R20 **a** Optical and Confocal Laser Scanning Microscopy images of PG channels after 6000 abrasion cycles at stretching strains of 0% and 80%. Scale bars: 200 μm . **b** Optical schematic of a

water droplet that cannot penetrate from the bottom to the top side.

Revision: To enhance the readability of Fig. 4e, in the revised manuscript, we simultaneously compared the unidirectional penetration time of the Janus membrane in the protective and open grooves and pores states for different abrasion cycles. The revisions to the original manuscript are marked **in red**.

Page 6, Line 179 – Line 181: “*Compared with the Janus membrane in open grooves and pores state with no protection, the water unidirectional penetration function fails after only 300 abrasion cycles, indicating the protection mode imparts Janus membrane outstanding mechanical stability.*” has been added.

Page 9, Line 248 – Line 250: “*When complete erosion of the microgrooves occurs after about 6000 abrasion cycles, namely $h \sim 0$, water droplets cannot penetrate from the bottom to the top side (Supplementary Fig. 14).*” has been added.

Corresponding Figure (*Fig. 4e*) has been rearranged for better readability.

Additional Figure (*Supplementary Fig. 14*) and *corresponding caption* have been added in the revised supplementary information.

5. Can authors explain the chemical durability of this fabricated Janus membrane? This aspect can enhance the quality of this manuscript.

Response: We thank the reviewer for this helpful question. Additional long-term chemical durability tests are conducted on the Janus membrane, as shown in Fig. R21.

Three stretched open membranes and three released membranes are directly immersed into different chemical solutions, i.e., HCl solution (pH \sim 4), NaOH solution (pH \sim 10) and 5 wt.% NaCl solution, for 10 days. After that, the protected membranes are restretched to the Janus mode at 80% strain, and the weight of collected water (WCW) is measured for 30 minutes for all the membranes to evaluate the stability. The results show that the membranes immersed in all the three chemical solutions generally possess the same WCW outputs. In addition, the fog collection capabilities of both the protected and open pore membranes generally remain consistent with the original state, as shown in Fig. 6d in the manuscript. These results indicate that the hydrophilic coating is resistant to corrosion in acidic, alkaline and salt environments.

Fig. R21 Comparison of the 30-minute WCWs of the Janus membranes in the protective and open states after 10 days of immersion in chemical solutions of HCl (pH ~ 4), NaOH (pH ~ 10) or 5 wt.% NaCl solution.

Notably, the core novelty of this work lies in the proposal of the mode switching strategy that can significantly enhance the mechanical durability of the Janus membrane regardless of the specific type or composition of the hydrophilic coating. Here, an alternative PVA/silica hydrophilic coating (*Adv. Mater.* **33**, 2102740 (2021)) is introduced for HCl and NaOH solution immersion tests instead of the Mesophilic-2000 coating. We can expect that this not only demonstrates the universal hydrophilic coating protection effect of the mode switching strategy, but also broadens the range of options for hydrophilic coatings that are both mechanically and chemically durable.

Revision: In accordance with the reviewer’s comments, we have added a discussion on the chemical stability of this Janus membrane to the revised manuscript. The revisions to the original manuscript are marked in red.

Page 11, Line 309 – Line 311: “*Considering the complexity of the practical application environment of the fog collector, the further long-term thermal, humidity and chemical stability performance of the Janus membrane in protection mode are also investigated.*” has been added.

Page 11, Line 315 – Line 325: “*In addition, membranes are kept in the conditions with different relative humidity (30%, 50%, 70% and 90%) and immersed into different chemical solutions (HCl, NaOH and NaCl). As shown in Fig. 6f-g, after 10 days’ treatment, the WCW of a period 30 minutes also keep consistent compared with the original state. The results indicate the outstanding thermal, humidity and chemical stability of the Janus membrane. What is worth noting that the core novelty of this work lies in the proposal of the mode switching strategy that can significantly enhance the mechanical durability of the Janus membrane regardless of the specific type or composition of the hydrophilic coating. Here, an alternative PVA/silica hydrophilic coating is introduced for HCl and NaOH solution immersion tests. We can expect that this not only demonstrates the universal hydrophilic coating protection effect of the mode switching strategy, but also broadens the range of options for hydrophilic coatings that are both mechanically and chemically durable.*” has been added.

Additional Figure (*Fig. 6g*) and *corresponding caption* have been added.

Finally, we sincerely appreciate the thoughtful and insightful comments. The manuscript has greatly benefited from these comments.

Reviewer #4:

The manuscript discusses a mode-switching strategy to protect Janus membranes through stretching and releasing. The work described is novel and emphasizes the long-term stability and mechanical durability of membranes, an area which is often overlooked in membrane development. However, the manuscript is incomplete and poorly structured in its current form. I suggest the manuscript be accepted after some revision.

1. To capture the membrane's ability for fog collection, the authors are advised to conduct thermogravimetric analysis to quantify water absorption and moisture content with varying strain %, and to discuss the factors that influence the same.

Response: We thank the reviewer for this helpful suggestion. Thermogravimetric (TG) analysis of the water absorption and moisture content with strain variation is conducted. Janus membranes with stretching strains of 0%, 20%, 40%, 60% and 80% are investigated after 30 minutes of fog collection. As shown in Fig. R22a, the TG analysis process starts at room temperature and continues to increase to 200°C in a N₂ atmosphere to quantify water absorption. All the samples with stretching strains ranging from 0% to 80% lost ~ 1.5% weight at 100°C (Fig. R22b), indicating that the Janus membrane possesses a limited moisture absorption capacity during the fog collection process because of the intrinsic hydrophobicity of silicone.

Fig. R22 a TG test results of samples with different stretching strains. **b** Variation of mass retention at 100°C for the Janus membrane at different stretching strains.

2. The ‘discussion’ section in its current form is not a discussion, but a conclusion.

Response: We thank the reviewer for pointing this out. The ‘Discussion’ section has been carefully polished and revised.

Revision:

Page 11, Line 327 – Line 333: “*In order to tackle the common and severe issue of Janus membrane function failure in real-world applications, an on-demand mode switching strategy is proposed in this*

study. The strategy of considering working and protection modes separately implemented by stretching and releasing the membrane. The stretching Janus mode enables water unidirectional penetration, and the releasing protection mode resists abrasion and impact.” has been rearranged.

Page 12, Line 338 – Line 346: *“In the proof-of-concept application, the Janus membrane serves as a fog collector, demonstrating its mechanical stability in harsh real-world environments. Our proposed strategy does not take into consideration the durability optimization of the hydrophilic coating. Improving the longevity of the hydrophilic coating would further enhance the robustness of the device, whether in protective mode or in working mode. Additionally, the wear resistance of the silicone has not been optimized. For instance, one improvement could involve incorporating carbon nanotubes into the silicone to enhance its wear resistance. In the future, a comprehensive enhancement of device durability could be achieved by combining a durable hydrophilic coating design with materials that offer both wear resistance and elasticity.”* has been added.

3. Much of the results' section also include methods of how measurements were taken, which should be limited to the methods section. As an example, lines 140-142 explain how the sandpaper test was carried out. This does not belong in 'results'. All methods must be moved to 'methods'.

Response: We thank the reviewer for these comments. The sentences describing the sandpaper test have been moved to the Methods section, and other similar sentences have also been rearranged in the revised manuscript.

Revision:

Page 12, Line 363 – Line 368: *“The hydrophilic reagent Mesophilic-2000 (hydrophilic PEG solution, MesoBioSys. Co. Ltd., Wuhan, China) was used in the sample surface hydrophilic modification. The reagent was applied on the stretched surface by cotton swabs, and the modified sample was placed at room temperature for 5 min to evaporate the solvent. The membrane was released and restretched three times for hydrophilic modification to obtain the hydrophilic top surface of the Janus membrane.”* has been rearranged.

Page 13, Line 387 – Line 391: *“The membrane in protection mode was mounted on the electrically controlled linear slide platform. A polymer probe covered with sandpaper was pressed against the membrane with a defined vertical pressure of 40 kPa. The long-time abrasion test was then performed by reciprocating the probe at a constant velocity of 50 mm/s.”* has been added.

4. As a result of the above, the 'methods' section is weak and lacks details needed to reproduce the results.

Response: Thank you for the comment. More experimental details have been added to the 'Methods' section in the revised manuscript.

Revision:

Page 13, Line 378 – Line 382: “*Due to the size of the stretching stage was larger than that of the SEM sample room, all the biaxially stretched membranes in experiments with structures were fixed on hard plastic plates by screws, and then the membranes were cut off for SEM characterization (EVO18, ZEISS, Germany). The FTIR spectra of the samples were recorded using a Nicolet iN10 Fourier Transform Infrared Microscope (Thermo Scientific Instrument Co. U.S.A).*” has been added.

Page 13, Line 397 – Line 408: “***Thermal, humidity and chemical stability tests*** *Janus membranes in protection mode were subjected to various conditions, including different temperature, humidity and chemical solutions, to investigate their long-term stability. After 10 days’ treatment, the membranes were stretched to Janus mode and the weight of collected water for 30 minutes was quantitatively compared with the original state (the freshly prepared Janus membranes) to evaluate the stability. Specifically, the thermal stability was tested by keeping the samples in refrigerator (-20°C), room condition (20°C) and oven (60°C). For humidity stability assessment, the membranes were exposed to homemade chambers with humidity levels of 30%, 50%, 70%, and 90%, respectively. In chemical stability tests, membranes were immersed into chemical solution of HCl (pH ~ 4), NaOH (pH ~ 10) and NaCl (5 wt.%), respectively. A hydrophilic coating, consisting of PVA/Silica solution¹⁰, was uniformly sprayed on the top side of Janus membrane by using a spray gun at a distance of 10 cm for chemical solution HCl and NaOH immersion tests.*” has been added.

Other similar sentences have also been carefully revised and rearranged in the revised manuscript.

Finally, we sincerely appreciate the thoughtful and insightful comments. The manuscript has greatly benefited from these comments.

REVIEWERS' COMMENTS

Reviewer #2 (Remarks to the Author):

The authors have done exceptionally well in satisfying all the queries of this reviewer. I thoroughly enjoyed reading the modified manuscript, especially the new experiments added and the discussion arising from them, for example, the temperature and humidity stabilities studies. In my opinion, these are more relevant for the application studied in this work. I suggest publication of this work in Nature Comm. in its current form.

Reviewer #3 (Remarks to the Author):

The authors have addressed all my comments/feedback. Therefore the revised manuscript can be accepted in the present format.

Reviewer #4 (Remarks to the Author):

Noteworthy results, as previously suggested in my comments. However, discussion still lacks substance, and is not suitable for publication in this journal.

Reply to the Reviewers' comments

Thank you for coordinating the review process for our manuscript, and we are deeply grateful for the editor's decision to publish our manuscript in principle. The reviewer's comments have been carefully considered. The manuscript has been carefully revised to address the reviewers' concerns. The major revisions are marked **in red** in the revised manuscript. The point-to-point responses to the comments are listed below.

Reviewer #2:

The authors have done exceptionally well in satisfying all the queries of this reviewer. I thoroughly enjoyed reading the modified manuscript, especially the new experiments added and the discussion arising from them, for example, the temperature and humidity stabilities studies. In my opinion, these are more relevant for the application studied in this work. I suggest publication of this work in Nature Comm. in its current form.

Response: We sincerely appreciate the reviewer for the professional and constructive comments. The overall quality of our study has greatly benefited during the last round of review.

Reviewer #3:

The authors have addressed all my comments/feedback. Therefore the revised manuscript can be accepted in the present format.

Response: We thank the reviewer for the invaluable comments and feedback.

Reviewer #4:

Noteworthy results, as previously suggested in my comments. However, discussion still lacks substance, and is not suitable for publication in this journal.

Response: We thank the reviewer for the constructive comments. The 'Discussion' section has been substantially polished and revised.

Revision: In accordance with the reviewer's comments, the "Discussion" section has been revised to include the discussions of technical advantages, potential limitations, and corresponding solutions of the mode switching strategy.

Page 10, Line 302 – Line 303: "*the durability of Janus membrane has been significantly enhanced, water unidirectional penetration could be reserved under harsh mechanical impacts.*" has been added.

Page 10, Line 304 – Line 306: "*water unidirectional penetration could be reserved even after 2000 abrasion cycles and 10 days of exposure to air. The sandpaper abrasion, finger rubbing, sand impact*

and tape peeling harsh durability tests further validate the mechanical stability and anti-clogging performance of the membrane.” has been deleted.

Page 11, Line 309 – Line 312: “*While the releasing protective mode can significantly enhance the durability of the Janus membrane in the face of external abrasion and impact, as well as during long-term storage, future work may focus on improving the durability of the flexible Janus membrane in its active working mode.*” has been added.

Page 11, Line 312 – Line 313: “*Our proposed strategy does not take into consideration the durability optimization of the hydrophilic coating.*” has been deleted.